



# Chemical analysis of the Asian Tropopause Aerosol Layer (ATAL) with emphasis on secondary aerosol particles using aircraft based in situ aerosol mass spectrometry

Oliver Appel[1,2], Franziska Köllner[2], Antonis Dragoneas[1,2], Andreas Hünig[1,2], Sergej Molleker[1], Hans Schlager[3], Christoph Mahnke[1,4], Ralf Weigel[2], Max Port[2,5], Christiane Schulz[1,6], Frank Drewnick[1], Bärbel Vogel[7], Fred Stroh[7], and Stephan Borrmann[1,2]

[1]Max Planck Institute for Chemistry, Mainz, Germany
[2]Institute for Atmospheric Physics, Johannes Gutenberg University, Mainz, Germany
[3]Deutsches Zentrum für Luft- und Raumfahrt (DLR), Institut für Physik der Atmosphäre, Oberpfaffenhofen, Germany
[4]now at: Institute of Energy and Climate Research (IEK-8), Forschungszentrum Jülich GmbH, Jülich, Germany
[5]now at: Montessori Zentrum Hofheim, Germany
[6]now at: Leibniz-Zentrum für Agrarlandschaftsforschung, Müncheberg, Germany
[7]Institute of Energy and Climate Research (IEK-7), Forschungszentrum Jülich, Jülich, Germany

**Correspondence:** Stephan Borrmann (stephan.borrmann@mpic.de)

**Abstract.** Aircraft borne in situ measurements of the chemical aerosol composition were conducted in the Asian Tropopause Aerosol layer (ATAL) over the Indian subcontinent in summer 2017 covering particle sizes below 3 μm. We have implemented a recently developed aerosol mass spectrometer, which adopts the laser desorption technique as well as the thermal desorption method for quantitative bulk information (i.e. a modified Aerodyne AMS), aboard the high altitude research aircraft M-55 *Geo-*

*physica*. The instrument was deployed in July and August 2017 during the StratoClim EU campaign (Stratospheric and upper tropospheric processes for better climate predictions) over Nepal, India, Bangladesh, and the Bay of Bengal, covering altitudes up to 20 km a.s.l. For particles with diameters between 10 nm and ~3 μm the vertical profiles of aerosol number densities from the eight research flights show significant enhancements in the altitude range of the ATAL. We observed enhancements in the mass concentrations of particulate nitrate, ammonium, and organics in a similar altitude range between approximately

13 km and 18 km (corresponding to 360 K and 410 K potential temperature). By means of the two aerosol mass spectrometry techniques, we show that the particles in the ATAL mainly consist of ammonium nitrate and organics. The single particle analysis from laser desorption and ionizaton mass spectrometry revealed that a significant particle fraction (up to 70 % of all analyzed particles by number) within the ATAL results from the conversion of inorganic and organic gas-phase precursors, rather than from the uplift of primary particles from below. This can be inferred from the fact that the majority of the particles

encountered in the ATAL consisted solely of secondary substances, namely an internal mixture of nitrate, ammonium, sulfate, and organic matter. These particles are externally mixed with particles containing primary components as well. The single particle analyses suggest that the organic matter within the ATAL and in the lower stratosphere (even above 420 K) can partly be identified as organosulfates, in particular glycolic acid sulfate, which are known as components indicative for secondary organic aerosol formation. Also, the secondary particles are smaller in size compared to those containing primary components

(mainly potassium, metals, and elemental carbon). The analysis of particulate organics with the thermal desorption method





shows that the degree of oxidation for particles observed in the ATAL is consistent with expectations about secondary organics that were subject to photochemical processing and ageing. We found that organic aerosol was less oxidized in lower regions of the ATAL ($< 380\,\mathrm{K}$) compared to higher altitudes (here 390 - 420 K). These results suggest that particles formed in the lower ATAL are uplifted by diabatic heating processes and thereby subject to extensive oxidative ageing. Thus, our observations are consistent with the concept of precursor gases being emitted from regional ground sources, subjected to rapid convective uplift, and followed by secondary particle formation and growth in the upper troposphere within the confinement of the Asian Monsoon Anticyclone (AMA). As a consequence the chemical composition of these particles largely differs from the aerosol in the lower stratospheric background and the Junge layer.

# 1 Introduction

The Asian Tropopause Aerosol Layer (ATAL) develops every year inside the Asian Monsoon Anticyclone (AMA) during the summer monsoon period (i.e. between June and September) at altitudes between approximately $14\,\mathrm{km}$ and $18\,\mathrm{km}$ (Vernier et al., 2009, 2011, 2015, 2018; Thomason and Vernier, 2013; Yu et al., 2017; Brunamonti et al., 2018; Zhang et al., 2019). The geographical extent of the AMA can reach from the Eastern Mediterranean to Western China, and correspondingly the ATAL is spatially, temporally, and seasonally highly variable (Yuan et al., 2019; Hanumanthu et al., 2020). Due to internal dynamical variability, the location and the shape of the anticyclone change from day to day and oscillate between a state with one anticyclone and a different state with two separated anticyclones (so-called modes). The modes often are referred to as western (Iranian) and eastern (Tibetan) mode (e.g. Zhang et al., 2002; Vogel et al., 2015; Nützel et al., 2016) and they affect the spatial distribution of the ATAL as well. Widespread within the Asian monsoon region and its surroundings, there are numerous ground sources of natural emissions and anthropogenic pollution (e.g. Lawrence and Lelieveld, 2010). As the summer monsoon is accompanied by large-scale convection especially in the Himalayan region and the Tibetan plateau, strong upward transport carries trace gases, aerosols, and precursor gases of natural and anthropogenic origin from the boundary layer aloft to the altitudes of the AMA (approximately 12 to $20\,\mathrm{km}$ a.s.l.), and in parts beyond into the lower stratosphere (Randel et al., 2010; Pan et al., 2016; Vogel et al., 2019). Soluble substances and aerosol particles experience scavenging and chemical processing during the transport through convective mixed phase clouds. As a consequence, the chemical composition and abundance of materials which are amenable for the formation of the ATAL are altered and thus differ from the initial boundary layer conditions (Barth et al., 2001, 2007a, b; Tost et al., 2010; Bela et al., 2016; von Blohn et al., 2011, 2013; Jost et al., 2017; Lawrence and Lelieveld, 2010). These constituents reach the cloud detrainment zone between approximately 8 and $16\,\mathrm{km}$ a.s.l. and accumulate inside the AMA together with other pollution components which arrive here in parts unaffected by cloud processing. Above the convective outflow the released emissions are rapidly oxidized by OH. The high abundance of $NO_x$ especially from lightning enhances the recycling of OH radicals, leading to an increased oxidative capacity of the air in the anticyclone (Lelieveld et al., 2018). The described processes influence the chemical composition of the air circulating inside the AMA throughout its vertical profile (Randel and Park, 2006; Park et al., 2009; Randel and Jensen, 2013; Vogel





et al., 2015; Pan et al., 2016; Bucci et al., 2020) and provide precursor gases for secondary aerosol formation. Such precursor
gases reach the upper troposphere (e.g. Höpfner et al., 2016, 2019; Fairlie et al., 2020), where homogeneous nucleation and
subsequent particle growth processes form secondary particles in clear air as well as inside the outflows of convective cloud
systems (Weigel et al., 2011, 2021a, b; Williamson et al., 2019). The exact details of the precursor gas mixture and the involved
processes for inorganic homogeneous nucleation (including $H_2SO_4$, $NH_3$, $HNO_3$ among others), as well as secondary organic
aerosol (SOA) formation still are unknown. However, mainly modeling studies show that the relevant aerosol components
within the AMA are nitrates, ammonium, sulfates, as well as organics, mineral dust, black carbon (Fadnavis et al., 2013;
Neely III et al., 2014; Yu et al., 2015; Gu et al., 2016; Lelieveld et al., 2018; Lau et al., 2018; Ma et al., 2019), some metals,
and to a small extent meteoric dust as well (Schneider et al., 2021). In this context ammonium nitrate (AN) is of particular
importance. As already pointed out by Metzger et al. (2002), the partitioning between the gas phase $HNO_3$ and $NH_3$ and
its condensed phase $NH_4NO_3$ influences the atmospheric life time of these substances, since ammonium nitrate has a much
longer residence time than its precursor gases. The authors concluded from their global model calculations (albeit with large
uncertainties and without considering seasonal cycles) that an extended ammonium nitrate aerosol plume should exist in the
upper troposphere at atmospheric pressure of 200 to 300 hPa (in a region characterized by efficient upward transport) from
South Eastern Asia to Africa as result of ammonia ground emissions (mostly from agriculture) in India and Southern Asia.
This was corroborated by satellite remote sensing as early as 1997 from the CRISTA instrument (Höpfner et al., 2019), and by
laboratory studies on the ammonia – nitric acid system (Wang et al., 2020). Further, Gu et al. (2016) concluded from GEOS-
CHEM 3D model simulations that particulate nitrate in the PM2.5 range is a dominant component of the upper troposphere and
lower stratosphere (UTLS) in the AMA region although it is not a major contributor to the boundary layer aerosol. According
to their calculations, the nitrate fractions of the PM2.5 aerosol can be higher than 60% at the 100 hPa level while being below 5
– 25% near the surface. Balloon based in situ data reported by Vernier et al. (2018) provided first indications for the presence of
nitrate aerosol in the ATAL. Finally, within the StratoClim project, remote sensing measurements from the aircraft borne (here
the Russian M-55 *Geophysica* high altitude research aircraft) near infrared spectrometer GLORIA by Höpfner et al. (2019),
gave clear evidence for solid ammonium nitrate as a major component of the ATAL inside the 2017 AMA. In their study also
first quantitative in situ measurements by means of our aerosol mass spectrometry were introduced, which unambiguously
demonstrated the existence of a nitrate aerosol layer in the ATAL altitude range. Desert dust may also be a component of the
regional UTLS aerosol and the ATAL. This was concluded from model simulations (Fadnavis et al., 2013; Lau et al., 2018;
Yuan et al., 2019), although measurements and experimental quantification still are missing.

Aerosol size distribution measurements were concurrently performed on the M-55 *Geophysica* during StratoClim inside of the
2017 AMA by Mahnke et al. (2021), using optical particle counters (UHSAS-A and NIXE-CAS) for the size range of 65 nm
to 3 µm in diameter. This range extends towards smaller particle sizes beyond the previous measurements by Yu et al. (2017),
who adopted a balloon borne POPS optical particle counter with a lower size detection limit of 140 nm. As detailed in Mahnke
et al. (2021), additional data from the COPAS high altitude condensation particle counter (Weigel et al., 2009, 2021a) allowed
adding one broad size bin extending from 10 nm to 65 nm to size distributions delivered by the UHSAS-A. By means of these
composite size distributions from the three instruments (i.e. COPAS, UHSAS-A, and NIXE-CAS) covering 10 nm to ~3 µm



aerosol back scattering ratios then could be derived and compared with the CALIOP satellite measurements of roughly the same time period during StratoClim. The results from both methods agree well and exhibit an aerosol layer between 14 km and 18 km altitude inside the AMA (Mahnke et al., 2021). Thus, with respect to number density, the ATAL in 2017 consisted mainly of particles with diameters below 3 $\mu$m with a broad peak in the size distribution around 100 nm with the highest number densities near 40 000 cm$^{-3}$ (Mahnke et al., 2021; Weigel et al., 2021a). Previous balloon borne observations also had indicated a maximum in the size distribution at diameters of around 100 nm (Yu et al., 2017; Vernier et al., 2018).

Diabatic vertical transport of air masses carries the aerosol particles of the ATAL aloft with slow (~1 – 1.5 K per day) vertical ascent rates in an anticyclonic large scale spiral movement (Vogel et al., 2019). These air masses cross the lapse rate tropopause which here in the tropics does not act as a transport barrier. Between approximately 380 K and 460 K of potential temperature isentropic mixing and "spiral stirring" with air masses from the neighboring regions occur. As a consequence of the arguments laid out for trace gases by Vogel et al. (2019), a probably small fraction of the contained aerosol can as well reach the tropical pipe and subsequently become globally distributed. Most likely, however, a larger part of the aerosol originating from the AMA is distributed together with other tropospheric constituents in the northern hemispheric extratropical lowermost stratosphere via quasi-isentropic transport (see e.g. Garny and Randel, 2016; Ploeger et al., 2017; Yu et al., 2017; Fadnavis et al., 2018). Since "eddy shedding" occurs a few times each summer (e.g. Popovic and Plumb, 2001; Vogel et al., 2014; Pan et al., 2016) parts of the accumulated aerosol particles and gases become amenable to distribution in the upper troposphere and for long range transport (see e.g. Lawrence and Lelieveld, 2010; Vogel et al., 2016; Fujiwara et al., 2021). As suggested by Yu et al. (2017), as much as 15 % of the northern hemispheric stratospheric aerosol surface area could originate in the AMA on an annual basis, albeit with considerable seasonal variability (Yuan et al., 2019). Whether or not the aerosol nucleated in the AMA is a source for the global stratospheric Junge layer is a still open and debated question. Some fraction of the freshly nucleated and aged particles inside the AMA are involved in ice cloud formation below the tropopause and in the tropical transition layer (Ueyama et al., 2018) as has been demonstrated by laboratory experiments for the case of ammonium nitrate aerosol (Wagner et al., 2020). Others may act as cloud nuclei further down in the tropical middle troposphere, which was suggested by Andreae et al. (2018) based on their measurements over the Amazon rain forest. For the higher altitudes the effects on the radiative budget and possibly heterogeneous chemistry (see e.g. von Hobe et al., 2011; Talukdar et al., 2012) are of concern. In general, aerosols in the UTLS are of relevance for the climate since the ice nucleation processes influence cirrus cloud properties (see e.g. Liu et al., 2009; Fadnavis et al., 2013) and thus their radiative effects. The extent of the regional radiative forcing at the top of the atmosphere in connection with the ATAL was estimated by Vernier et al. (2015) to be in the order of $-0.1$ W m$^{-2}$. Although considerable uncertainty still exists concerning the radiative forcing of the ATAL, from Ridley et al. (2014) it can be concluded that the radiative effects may be comparable to the global aerosol forcing (0.05 to 0.12 K) from moderate volcanic eruptions since 2000. Based on sensitivity studies using model simulations and satellite observations, Fadnavis et al. (2019) concluded that the regional upper tropospheric aerosol may aggravate the draughts over the Indian sub-continent during El Niño periods. In this study we provide detailed observational data on the chemical aerosol composition of the ATAL and its surroundings from in situ measurements carried out on board the Russian M-55 *Geophysica* high altitude research aircraft within the StratoClim project. While previous work has been focused on nitrate aerosol, we demonstrate here by means of quantitative and qualitative





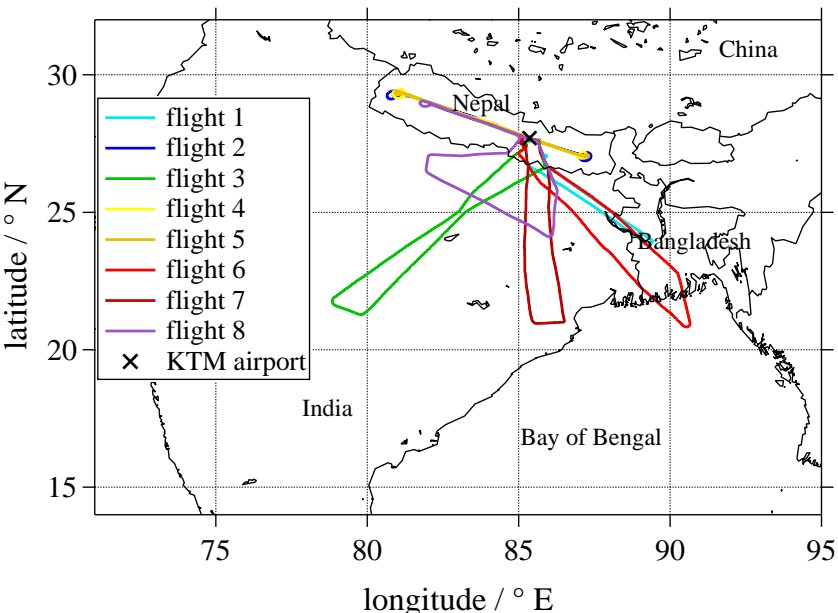

**Figure 1.** Paths of the eight research flights during the StratoClim field campaign 2017 over Nepal, India, Bangladesh, and the Bay of Bengal. The flight dates were Flight 1: 27 July, Flight 2: 29 July, Flight 3: 31 July, Flight 4: 2 August, Flight 5: 4 August, Flight 6: 6 August, Flight 7: 8 August, Flight 8: 10 August 2017.

aerosol mass spectrometry that the ATAL not only contains ammonium nitrate particles but also a significant fraction of
particulate organics, the properties of which also are subject of this study. Furthermore, the presented measurements include results for particulate sulfate within the AMA.

## 2    The StratoClim field campaign, measurement platform, and instrumentation

The measurements were taken in the framework of the StratoClim (see: http://www.stratoclim.org/, accessed on 27 Jan 2022) EU project with the goal to better understand the meteorological and chemical processes of the AMA. The base for the field
campaign was the Tribhuvan International Airport (KTM) in Kathmandu, Nepal, where eight research flights were performed between 27 July and 10 August 2017. The adopted platform was the single-seated Russian high altitude research aircraft M-55 *Geophysica*, which can carry a payload of up to 2000 kg to a nominal ceiling of 21 km. The eight research flights had a duration between 2 h 42 min and 4 h 28 min and the highest GPS altitude reached during StratoClim was 20.0 km. Fig. 1 shows the flight paths for the StratoClim aircraft campaign 2017 over Nepal, India, Bangladesh, and the Bay of Bengal.

**In situ aerosol mass spectrometry:** For in situ measurements of the aerosol chemical composition the newly developed ERICA instrument (ERC Instrument for the Chemical composition of Aerosols) was implemented on the M-55 *Geophysica*

(Hünig et al., 2021; Dragoneas et al., 2022). The two commonly adopted methods for in situ real-time mass spectrometric analysis of aerosol particles are (1) laser desorption ionization (LDI) of particles followed by time-of-flight mass spectrometry (Suess and Prather, 1999), and (2) thermal desorption with subsequent electron impact ionization (TDI) as implemented in the Aerodyne AMS (Drewnick et al., 2005; Canagaratna et al., 2007). The ERICA combines these two techniques in a single, fully automatically operating apparatus. The instrument has been described in detail by Hünig et al. (2021) and Dragoneas et al. (2022) and is only briefly reviewed here:

Ambient sample air is supplied to the ERICA by a specially designed shrouded inlet (Dragoneas et al., 2022), which consists of a two-staged diffusor that reduces the required sampling line velocity for isokinetic sampling by a total factor of 39. The principal design is based on the aerosol inlet used in the IAGOS-CARIBIC project (Brenninkmeijer et al., 2007) and for the ERICA the inlet design and its performance has been simulated by computational flow dynamics. Before entering the mass spectrometer the sample air passes through a pressure-regulating critical orifice (Molleker et al., 2020) which ensures the reliable performance of the aerodynamic lens (Peck et al., 2016) at the entrance of the ERICA. The critical orifice maintains a constant mass flow or normal volumetric flow (referenced to normal temperature and pressure NTP: 20 °C and 1013.25 hPa) of 1.45 $cm^3\,s^{-1}$ into the instrument. The aerodynamic lens is designed to be operated at a pressure of 4.53 hPa, provided by a critical orifice with a diameter of ~100 µm at ground level, which increases to ~400 µm at lower stratospheric ambient pressures of 60 hPa.

The ERICA-LAMS (ERICA laser ablation mass spectrometer) unit implements the LDI technique. Two continuous wave lasers detect and classify the individual particles by size and are used to trigger a pulsed UV desorption/ionization laser (Hünig et al., 2021), similar to, e.g. the ALABAMA (Brands et al., 2011) or the PALMS (Murphy, 2007) instruments. The vacuum aerodynamic diameter of the particles ($d_{va}$) is derived from the particle velocity by applying a calibration with particles of known diameter, density, and shape (here, monodisperse polystyrene latex particles in a diameter range between 80 nm and 5.1 µm (Hünig et al., 2021)). The ions generated by the UV laser pulse (~4 mJ at a wavelength of 266 nm) are extracted by switched electrical fields to generate positive and negative ion mass spectra for each desorbed aerosol particle. Due to a relatively long idle time of >120 ms of the desorption laser, most particles continue flying unaffected downstream in the vacuum chamber towards the ERICA-AMS unit. They hit a tungsten vaporizer, where the non-refractory components are vaporized at around 600 °C. The vapour is ionized by means of the electron impact method and a CTOF-MS (compact time-of-flight mass spectrometer) is used to generate uni-polar positive mass spectra of small particle ensembles.

The particle size range detectable by the ERICA-AMS is given by the transmission efficiency of the aerodynamic lens (vacuum aerodynamic diameter of the particles between 110 nm and 3.5 µm, Xu et al. (2017)). Although Hünig et al. (2021) indicates an even lower cutoff of less than 90 nm, the smallest particles will not significantly contribute to the mass concentrations measured by the ERICA-AMS during StratoClim. The lower cutoff diameter of the LDI unit is limited by the optical particle detection efficiency to 180 nm. The short passage through the inlet and tubing does not significantly alter the detectable size range and the losses have been estimated to be below 20 % for particles up to 3 µm even at low ambient pressures of 65 hPa (Dragoneas et al., 2022).

The data of the ERICA-AMS are processed with TofWare 2.5.7 (Tofwerk) as detailed in the Supplement S1.1 and S1.2. The





detection limits of the mass concentrations are calculated from the background measurement with closed shutter as in Drewnick et al. (2009), however in this publication we analyze the background noise using a Savitzky-Golay (SG) filter (Savitzky and Golay, 1964). The method is further described in Supplement S1.3 and is similar to those adopted by Reitz (2011) and Schulz et al. (2018). During the StratoClim aircraft campaign 2017 we found an average detection limit of $0.12\,\mu g\,m^{-3}$ for nitrate, $0.13\,\mu g\,m^{-3}$ for sulfate, $0.50\,\mu g\,m^{-3}$ for organics, and $0.73\,\mu g\,m^{-3}$ for ammonium, each representative for a sampling period of $10\,s$, i.e. one measurement cycle consisting of aerosol and background measurement. For longer averaging times $t$ (e.g. the total residence time in a certain altitude or $\theta$ bin) the detection limit scales with $1/\sqrt{t}$. The detection limits are comparable to values determined with the common filter method during ground measurements as shown by Hünig et al. (2021).

Single-particle mass spectra were obtained by the ERICA-LAMS. Considering only spectra sampled outside of clouds a total of 109 453 spectra were analyzed from the StratoClim data set; 99 % of the spectra include size information and 92 % of the spectra have dual polarity. We used the software package CRISP (Concise Retrieval of Information from Single Particles; Klimach, 2012) to perform $m/z$ (ion mass to charge ratio) calibration, peak area integration, and particle classification. Particles were classified by a combination of fuzzy-c means clustering (Hinz et al., 1999; Roth et al., 2016; Schneider et al., 2021) and the marker ion method (Köllner et al., 2017, 2021). For this study, we classified particles in two main categories: particles containing solely components typical for secondary formation like nitrate, sulphate, and organics (secondary type) and particles containing a mixture of secondary as well as primary components like soot, metals, salt, and organics from biomass-burning (primary or mixed type). Details on the classification methods of the particles can be found in the Supplement Sect. S2.1. As a next step, the fraction and scaled number concentrations of the particle types are determined. For the calculation of particle fractions, we summed up total analyzed particles (i.e. detected, desorbed, and ionized) and particles from a specific type in certain bins of variables (here, altitude, potential temperature, and $d_{va}$). For each bin, we divided the number of particles of a specific type by the number of all analyzed particles, providing the particle number fraction ($PF$). Further, by multiplying with the measured particle number concentration at the first detection stage ($N_0$) of the ERICA, we acquire the scaled number concentration of a specific particle type ($PF \cdot N_0$). This scaling method assumes that the hit rate of the ERICA-LAMS is independent on particle characteristics, such as size, composition, and shape. However, earlier studies show that this is not the case (Thomson et al., 1997; Kane et al., 2001; Moffet and Prather, 2009; Brands et al., 2011). As a result, we only discuss the variation in the scaled number concentrations as a function of other parameters (here: altitude and potential temperature), rather than discussing the absolute values of number concentrations.

**Aerosol size distribution and cloud particle measurements:** A modified UHSAS-A optical particle counter covering a nominal particle diameter from $65\,nm$ to $1\,\mu m$ was deployed on the aircraft as an underwing probe for aerosol size distribution measurements. As described in more detail by Mahnke et al. (2021), for the operation of the UHSAS-A at atmospheric pressures below $130\,hPa$ several modifications, e.g. of the flow system, had to be implemented to ensure reliable performance under the ambient conditions encountered during StratoClim.

A CDP optical wing sonde (Cloud Droplet Probe from Droplet Measurement Technologies, Longmont, CO, USA) with a diameter range from $2.5\,\mu m$ to $46\,\mu m$ was used to detect the presence of cloud and ice particles. In this study the CDP has been




used to identify cloud passages (i.e. periods with ice water contents above $0.1\,\mathrm{mg\,m^{-3}}$, averaged over 11 seconds) and exclude single particle spectra and ERICA-AMS data during the passages from our analysis to avoid potential artefacts from the inlet
system like droplet impact or ice shattering.

The number concentrations of aerosols with sizes down to nucleation mode were determined with the four-channel flow-through condensation particle counter COPAS (COndensation PArticle counting System). COPAS is operated with a chlorofluorocarbon (FC-43) as the working fluid, its particle detection and data storage occur with a temporal resolution of maximum 1 Hz. Three COPAS channels operate with 50 % detection particle diameters $d_{p,50}$ of $6\,\mathrm{nm}$, $10\,\mathrm{nm}$ and $15\,\mathrm{nm}$, respectively.
The fourth COPAS channel (with $d_{p,50} = 10\,\mathrm{nm}$) detects particles after their exposure to temperatures of $270\,°\mathrm{C}$ by means of a heated sample flow line. In this way particle concentrations of non-volatile (nv) or refractory particles (e.g. soot, mineral dust, metallic aerosol material, etc.) are determined. The counting performance of COPAS under variable pressure conditions is described by Weigel et al. (2009), and its capabilities have been demonstrated in several deployments of COPAS associated with the M-55 *Geophysica* (see Borrmann et al., 2010; Weigel et al., 2011, 2021a, b, ,and references therein)

**Other instruments and methods:** Ambient temperature $T$ and pressure $p$ as well as coordinates (GPS altitude, latitude, longitude), and aircraft specific data like roll and pitch angle, were supplied by the UCSE unit (Unit for Connection with the Scientific Equipment; Sokolov and Lepuchov, 1998) from the aircraft's avionic system. The accuracies for the 1 Hz-resolved UCSE data are $\pm 1\,\mathrm{hPa}$ and $\pm 2\,\mathrm{K}$ for the ambient pressures and temperatures, respectively. The potential temperature $\theta$ has been
calculated from the aircraft data acoording to $\theta = T \cdot \left(\frac{1000\,\mathrm{hPa}}{p}\right)^{\frac{R_L}{c_p}}$ with the specific gas constant $R_L = 287.058\,\mathrm{J\,kg^{-1}\,K^{-1}}$ and specific heat capacity $c_p = 1005\,\mathrm{J\,kg^{-1}\,K^{-1}}$.

For measurements of NO with SIOUX a chemiluminescence technique is adopted. Here, a cooled photomultiplier detects the light emission from the reaction of NO with ozone ($O_3$). $O_3$ is produced in the instrument by a dielectric barrier discharge. The detector is identical to the one used for measurements from an in-service aircraft during more than 460 flights since
2005 (CARIBIC Project) and is described in detail by Stratmann et al. (2016). The precision and accuracy of the SIOUX NO measurements are 7 % and 10 %, respectively. The NO detection limit is 3 pptv.

## 3   Results and discussion

### 3.1   Chemical composition of the ATAL aerosol particles as measured by the ERICA-AMS

The ERICA-AMS provides mass concentrations of particulate organics, nitrate, sulfate, and ammonium. Figure 2 shows the
median mass concentrations as function of altitude and potential temperature ($\theta$) together with the $25th$ and $75th$ percentiles. The profiles of the eight individual research flights are provided in the Supplement S3.1. Based on the distribution of our in situ mass concentration measurements, several atmospheric layers can be identified:

1. The boundary layer (BL) with increased mass concentrations below $320\,\mathrm{K}$,

2. The free troposphere (FT) between $320\,\mathrm{K}$ and $350\,\mathrm{K}$ with low values for the mass concentrations ($< 0.5\,\mathrm{\mu g\,m^{-3}}$)





3. The ATAL between 355 K and 420 K. This is comparable to the altitude range of the ATAL given in Vernier et al. (2011) of 13 to 18 km (corresponding to ~360 - 410 K during StratoClim) detected by the satellite based Lidar instrument CALIOP (Cloud-Aerosol Lidar with Orthogonal Polarization).

4. Above 420 K mostly sulfate aerosol is detected which is a typical characteristic of the lower stratosphere (LS).

According to European Centre of Medium-Range Weather Forecasts (ECMWF), ERA-Interim reanalysis data for the time period of the research flights, the lapse rate tropopause was between 369 K and 396 K of potential temperature with a mean level of 380 K. Thus, the aerosol layer visible in the mass concentration data extends well into the LS. The top of the AMA system confinement in 2017 was found at roughly 420 K by means of in situ trace gas measurements (von Hobe et al., 2021) on the M-55 *Geophysica*. This level of potential temperature coincides with a significant decrease in the particle number concentrations towards stratospheric values as reported by the COPAS condensation particle counter for particles with diameters between 10 nm and 1 μm (Mahnke et al., 2021; Weigel et al., 2021a). Including deep convection the region of convective outflows extends from about 12 km to 17 km – approximately corresponding to $\theta$ levels between 360 to 385 K – with a maximum intensity around 14 km. Based on back-trajectory analyses and satellite observations, Bucci et al. (2020) concluded for the period of the StratoClim aircraft campaign 2017 that the age of air up to 15 km is rarely more than 5 days and that these young air masses dominate the air composition here. Bucci et al. (2020) also identified a transition layer between 15 km and 17 km with ages of air masses between one and two weeks. In this transition region convection still is the major contributor of air masses. Above 17 km, the mean age of the sampled air is ~20 days with the convective contribution decreasing to zero around 20 km. Thus, the ATAL layer discernible in Fig. 2 is situated in an altitude region with air of several weeks of age, which still is strongly influenced by deep convection. In this meteorological situation of the AMA in 2017, organic and inorganic precursors for secondary aerosol formation, as well as primary particles originating from below easily reach the altitudes of the ATAL. Consequently, the ATAL chemical composition is largely determined by the relative contributions of new particle formation and secondary particle growth at altitude compared to the upward transport of already nucleated secondary or of primary particles from below.

The vertical profiles in Fig. 2 show clear enhancements of aerosol mass concentrations in the UTLS, especially for particulate organics, nitrate, and ammonium. In the ATAL altitude range the vertical number density profiles of submicrometer sized particles, as concurrently measured by the COPAS and the UHSAS-A, also exhibit significantly elevated values with respect to the middle troposphere and the lower stratosphere (Mahnke et al., 2021; Weigel et al., 2021a). This is evident from Fig. 3, which is a comprehended representation of the data from their studies, and which shows the results of particle number concentration larger than 10 nm and larger than 65 nm as a function of altitude and potential temperature. In these publications and in Weigel et al. (2021b) it was demonstrated that new particle formation and subsequent particle growth by condensation and coagulation can be frequently observed in the lower ATAL region below the tropopause (between 355 and 370 K). Note however, that COPAS data for particles between 6 and 10 nm in diameter – indicative of such transient new particle formation events – are not included in Fig. 3. Thus, the ATAL in the 2017 AMA could be identified by all of our in situ aerosol particle instruments aboard the M-55 *Geophysica* as enhancements in the respective vertical profiles. Remarkably, the vertical profile of sulfate does

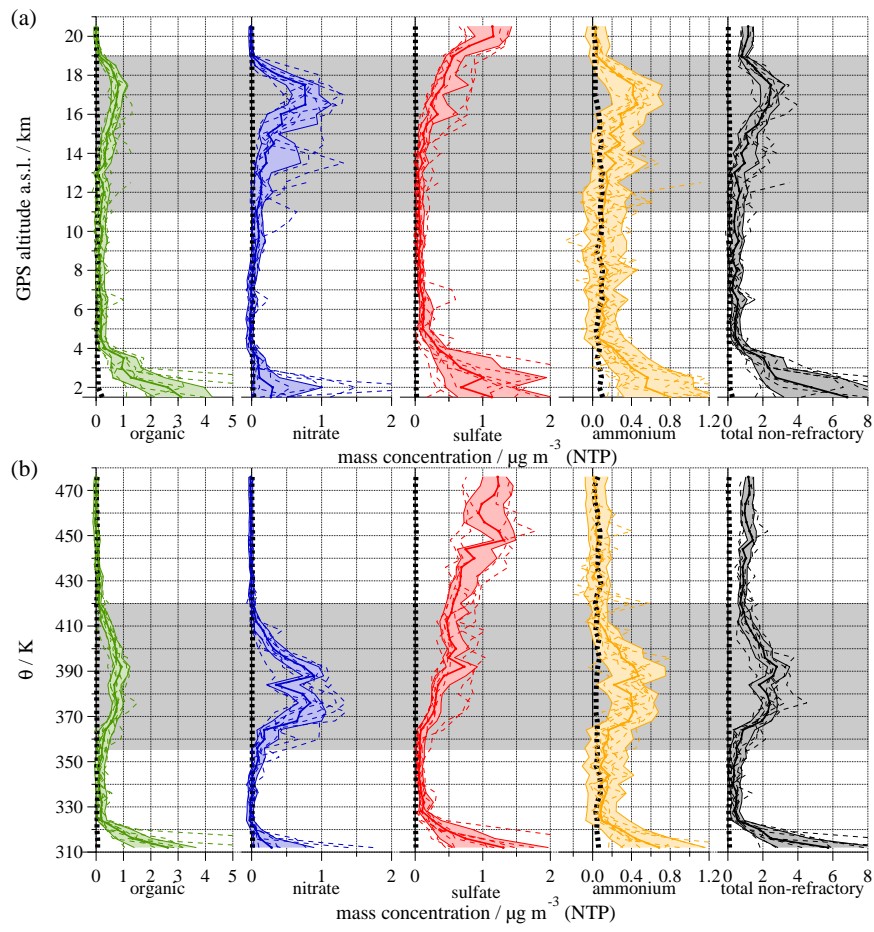

**Figure 2.** ERICA-AMS mass concentrations of particulate organics (green), nitrate (blue), sulfate (red), and ammonium (orange) as a function of (a) GPS altitude and (b) potential temperature $\theta$. The thick line and the shaded area represent the median and $25th/75th$ percentiles in the corresponding altitude or $\theta$ bin of the entire StratoClim aircraft campaign 2017. The dashed lines represent the median of each individual research flight. The right panel displays the sum of all species measured by the ERICA-AMS (black). The shaded background indicates the approximate altitude range of the ATAL. For each altitude bin the detection limit is displayed as a dotted line close to the ordinate.





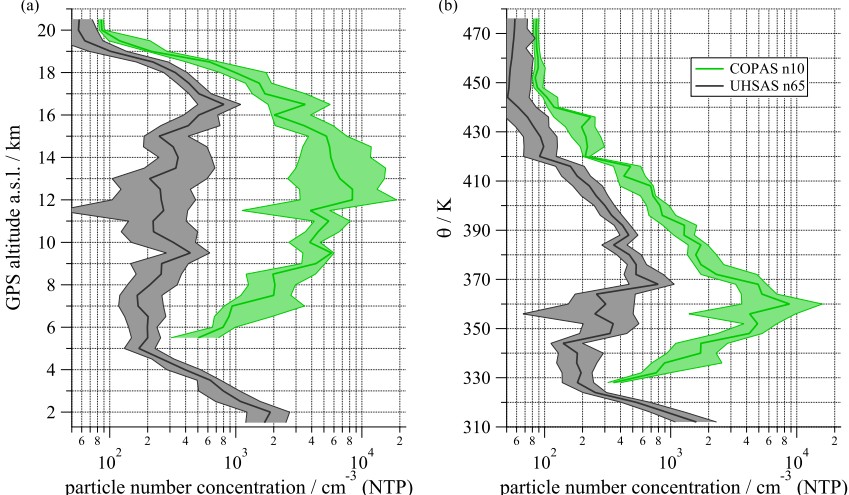

**Figure 3.** Vertical profile of the median submicron aerosol particle number concentrations as measured by the COPAS and the UHSAS-A (n10, n65: particles with diameters larger than 10 nm and 65 nm, respectively) as a function of (a) GPS altitude and (b) potential temperature $\theta$.

not show a peak in the ATAL region. It steadily increases towards the stratosphere becoming the dominant aerosol constituent
above 420 K. This is interesting, because several studies in the literature (e.g. Neely III et al., 2014; Yu et al., 2017; Vernier et al., 2015) hypothesized sulphur dioxide to be a main precursor for the ATAL layer formation, which is not unreasonable considering the high level of regional anthropogenic $SO_2$ emissions. However, for the particles in the ATAL of the AMA 2017, nitrate and organics are the major constituents. The vertical transport above 360 K, across the tropopause and up to ~460 K is driven by diabatic heating in an upward spiralling anticyclonic motion (Vogel et al., 2019). The observed increase in sulfate
mass concentration is inside this upward spiralling range with growing influence of the stratospheric background aerosol at the higher altitudes, because lower stratospheric air increasingly mixes with the upward transported AMA air masses (Vogel et al., 2019).

In terms of absolute values the ERICA-AMS measurements show increased total mass concentrations in the altitude range of the ATAL (grey shaded area in Fig. 2) with a maximum of more than $2 \, \mu g \, m^{-3}$(NTP). We found detectable concentrations
of particulate nitrate up to 19 km a.s.l. or 420 K in potential temperature by means of the ERICA-AMS, whereas particulate organics even reach 19.5 km or 430 K. Thus, regarding the main particle components we can determine the upper boundary of the ATAL to be higher than 18 km, which is the upper boundary as detected by satellite observations (Vernier et al., 2011). Figure S1 in the Supplement demonstrates the average potential temperature and vertical spread-out of the organics profile to be larger than for the nitrate layer. At the lower boundary of the ATAL the nitrate signal shows an increase with altitude at
8 km, when averaging over the whole campaign. However, during some flights no nitrate signal was detected below 13.5 km. Overall the concentrations and vertical distributions varied strongly from day to day (dashed lines in Fig. 2 and Fig. S2-S9 in the supplement). While this can in general be explained by different flight paths, Hanumanthu et al. (2020) found similar





day-to-day variations for the total mass concentration detected over the foothills near Nainital, India during a balloon campaign in August 2016. Nitrate contributes about 30 % to the non-refractory mass of the ATAL. This is more than common fractions

of nitrate obtained during boundary layer measurements in Asia (Zhou et al., 2020) or other parts of the world (Jimenez et al., 2009), even though the occurrence of nitrate aerosol is generally associated with agricultural emissions in the boundary layer (e.g. Hock et al., 2008). Together with the sufficiently available ammonium (see Section 3.2), it can be assumed that the nitrate found in the ATAL is predominantly existent in the form of ammonium nitrate (AN). A first quantification of particulate AN concentrations during StratoClim was obtained by Höpfner et al. (2019), who used the limb sounding instrument GLORIA

as well as data from the ERICA-AMS. The concentrations measured by the two instruments show a good agreement between the instruments, which operate on very different physical principles. Concerning the particle phase, they concluded that the AN particles are solid based on their GLORIA measurements, dedicated AIDA chamber laboratory studies, and single particle aerosol mass spectra delivered by the ERICA-LAMS. The high abundance of AN is a distinguishing property of the ATAL aerosol. The median concentration of particulate nitrate shows a maximum of about $0.8\,\mu g\,m^{-3}(NTP)$ around $17\,km$ or 370-

$390\,K$ during the StratoClim aircraft campaign 2017. Höpfner et al. (2019) emphasized the importance of AN, but not only nitrate is enriched here. Organics – which are not detected by the GLORIA instrument – contribute similar amounts to the particulate mass concentration and may play a similar role for the ATAL. One could even imagine a very faint "organic ATAL" without AN, which could have formed in periods with low emissions of ammonia from agricultural activities.

As the AMA supports an accumulation of precursor gases and aerosol particles, it is worthwhile investigating the spatial

distribution of the measured aerosol chemical properties. A suitable parameter for this is the equivalent latitude (EQL) as introduced by Ploeger et al. (2015), which defines the distance from the centre of the AMA based on gradients of potential vorticity. The value of 90° for EQL designates the centre of the AMA with decreasing values indicating growing distance. For Fig. 4 we binned averages of the measured organic and nitrate mass concentrations with respect to EQL (2° per bin) and potential temperature $\theta$ (5 K per bin). Figure 4b) shows, that the highest concentrations of nitrate appear in the central region

of the monsoon anticyclone. The mass concentration of particulate organics shows a broader distribution in vertical as well as horizontal extension (Fig. 4a)). However, these results represent only a brief period of the AMA and the underlying dynamics of the anticyclone may well lead to very different distributions in the course of time. Nevertheless, the data show that the ATAL for the time of StratoClim was not only confined in the vertical direction but also indicates a decrease towards the edge of the AMA in the horizontal distribution.

## 3.2 Acidity of the ATAL aerosol

In the Asian monsoon region $NO_x$ is likely to be abundant due to anthropogenic emissions as well as from lightning (Tost, 2017; Fairlie et al., 2020; Metzger et al., 2002; Stratmann et al., 2021), reacting with water to form $HNO_3$. In fact, in situ measurements of NO during Stratoclim showed enhanced concentrations in the ATAL with volume mixing ratios above 0.5 ppbv at altitudes between 16 and 18 km (Stratmann et al., 2021). This is demonstrated in Fig. 5 where the particulate nitrate mass

concentration is plotted against the gas phase NO mixing ratio, colour coded with potential temperature $\theta$. Between ~360 K and 400 K NO is increasing with altitude and with particulate nitrate mass (lower regression line in Fig. 5) since this altitude





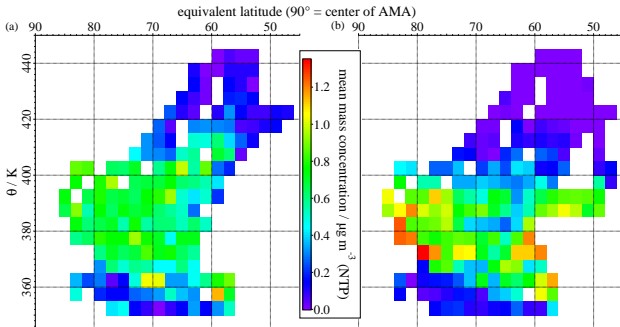

**Figure 4.** Curtain plot of (a) organic and (b) nitrate mass concentration as a function of $\theta$ and equivalent latitude (AMA-centred). The values for organics are based on the data from the flights 1, and 4 through 8.

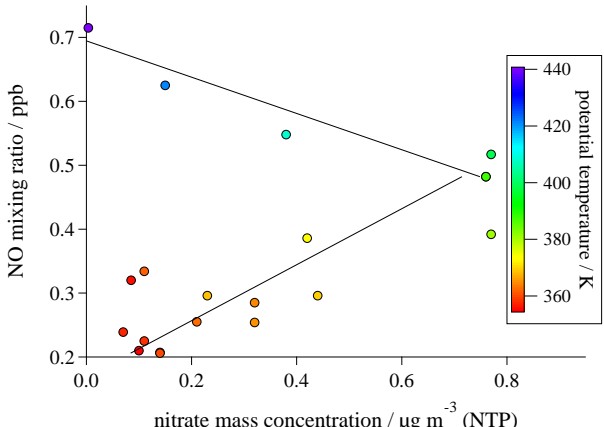

**Figure 5.** Nitrate mass concentration measured by the ERICA-AMS plotted against NO mixing ratio of SIOUX. The regression lines are shown to indicate the trends and are based on values for altitudes 10-17 km and 17-19 km, respectively.

band roughly corresponds to the outflow region of the convective clouds with lightning activity. Above ~400 K and in the lower stratosphere NO increases further as result of the photolysis of nitrous oxide ($N_2O$) while the particulate nitrate mass declines (upper regression line in Fig. 5). The formation of particulate nitrate usually requires nitric acid to find a reaction partner to

form a less volatile substance. The capability of the gas phase constituents ammonia and nitric acid for homogeneous nucleation of new particles consisting of ammonium nitrate (followed by fast post-nucleation growth) has been demonstrated and studied in dedicated laboratory experiments by Wang et al. (2020). Even though ammonia is soluble in water and can thus be scavenged by precipitation during convective uplift, a low retention coefficient in the ice phase (Hoog et al., 2007; Jost et al., 2017; Ge et al., 2018) can play a role for ammonia reaching altitudes of cumulonimbus outflows in significant quan-

tities especially during the Asian monsoon with prevalent deep convective clouds. Höpfner et al. (2016, 2019) demonstrated the occurrence of ammonia from agricultural sources above the convective outflow and also the influence of ammonia to the



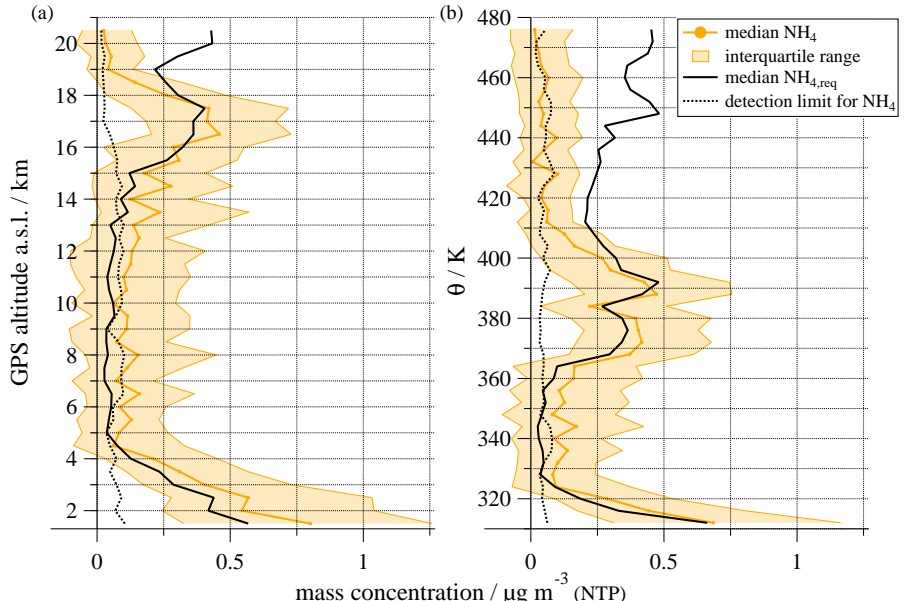

**Figure 6.** Vertical profile of the mass concentration of particulate ammonium (orange) with median and interquartile ranges measured by the ERICA-AMS as well as the median calculated ammonium required for full neutralization of the inorganic components ($NH_{4,req}$, black line) as a function of (a) GPS altitude and (b) potential temperature $\theta$. The shaded areas delineate the $25th$ and $75th$ percentiles, and the detection limit of ammonium is shown as a dotted line.

formation of ammonium nitrate in the ATAL.

The stoichiometric ratio of nitrate and sulfate on one hand, and ammonium, which can neutralize these species, on the other hand provides an indication on the acidity of the aerosol (Zhang et al., 2007). The mass concentration of ammonium can

directly be measured with the ERICA-AMS, while the concentration of nitrate ($[NO_3]$) and sulfate ($[SO_4]$) determine the concentration of ammonium required for neutralization $[NH_{4,req}]$ according to:

$$[NH_{4,req}] = \frac{36}{96} \cdot [SO_4] + \frac{18}{62} \cdot [NO_3] \tag{1}$$

Figure 6 shows the vertical profile of $[NH_4]$ and $[NH_{4,req}]$ against altitude and potential temperature. The aerosol particles can be considered as fully neutralized, if the measured ammonium concentrations are in agreement with $[NH_{4,req}]$ in the

corresponding bin. However, the neutralization balance of organics could not be included in the analysis due to the lack of the required data. Particles up to altitudes of 18 km or a potential temperature of 400 K contain approximately sufficient ammonium to neutralize the nitrate and sulfate content. Up to 19 km or 410 K ammonium is still above the detection limit, but insufficient for the neutralization of the observed nitrate and sulfate. For higher levels, no particulate ammonium was detectable, in agreement with the observations of sulfuric acid in the stratosphere (e.g. Murphy et al., 2014). In essence, the

vertical transport of ammonia and/or ammonium reaches altitudes up to 19 km influencing the aerosol acidity.



### 3.3 Secondary organic and inorganic particles within the ATAL

In this section we discuss the role of secondary particle formation for the chemical composition and physical properties of the ATAL. The ERICA-AMS is not suited to detect refractory material like dust, metals and salt, which usually are indicative for the presence of primary aerosol particles. Non-refractory nitrate and sulfate as detected by the ERICA-AMS mostly represent

secondary aerosol, however it can not be differentiated between pure secondary particles or coatings of nitrate and sulfate on pre-existing primary particles. The complementary single particle analysis by the LDI technique (here the ERICA-LAMS) allows us to study the internal mixing state of individual particles present in the ATAL. The mean bipolar single particle mass spectra of the secondary and the primary or mixed particle types (definitions see Sect. 2) are provided in Figs. 7 and 8. To note, we averaged over the entire single particle data set during StratoClim 2017, excluding data inside clouds. The primary

type is characterized by the detection of primary species like potassium and elemental carbon in the BL and ATAL region, along with sodium, other minerals, and anthropogenic metals in the BL (Figs. 8b and 8c). In the stratosphere above $420\,\mathrm{K}$, primary particles are mainly composed of meteoric material (Fig. 8a; see also Schneider et al., 2021). However, as evident from the negative ion spectra, these particles are typically coated with secondary substances, such as sulfate and nitrate (Fig. 8). The identification of such mixing states requires the ability to record both ion polarities for each individual analyzed aerosol

particle. Particles identified as secondary type are characterized by an internal mixture of solely secondary species, namely nitrate ($m/z$ +30 ($\mathrm{NO^+}$) and -46 ($\mathrm{NO_2^-}$)), ammonium ($m/z$ +18 ($\mathrm{NH_4^+}$)), sulfate ($m/z$ -97/99 ($\mathrm{HSO_4^-}$)), and organic matter ($m/z$ +27 ($\mathrm{C_2H_3^+}$) and +29 ($\mathrm{C_2H_5^+}$), likely from alkyl fragments (Silva and Prather, 2000); see Fig. 7). Hence, the so defined secondary particle type is not internally mixed with primary particle components. Further, we separated the secondary particle type into type 1 and type 2. Type 1 is characterized by a dominant $\mathrm{NO^+}$ signal, whereas type 2 shows organic peaks in the

same order of magnitude as $\mathrm{NO^+}$ (Fig. 7). Figure 9 presents the occurrence of secondary particle types 1 and 2 as a function of altitude and potential temperature. This figure shows the cumulative particle fraction ($PF$, left panels) and the scaled number concentration ($PF \cdot N_0$, right panels) of the particle types (definition see Sect. 2). The size-resolved particle fraction of these particle types is given in Fig. 10. Note that in Fig. 10 the primary particle type is restricted to particles found in the ATAL, since boundary layer aerosol and meteoric dust are not in focus of this study.

We found that the secondary particle type 1 is predominantly abundant within the ATAL, with a maximum particle fraction of 70 % by number (Fig. 9). In contrast, in the BL and FT, the abundance of particle type 1 is low, while the particle type 2 shows maximum fraction and number concentration. With increasing altitude within the ATAL, the abundance of type 1 is increasing and reaches maximum values between 380 and $400\,\mathrm{K}$ (between 17 and $18\,\mathrm{km}$). Above $400\,\mathrm{K}$ potential temperature, the abundance of particle type 1 is again decreasing with $\theta$ and in the lower stratosphere (above $440\,\mathrm{K}$) this particle type is

not evident. Thus, the secondary particle type 1 (with the dominant $\mathrm{NO^+}$ ion peak in the cation spectrum) is the dominant particle type in the ATAL region, consistent with the high mass fraction of AN measured by the ERICA-AMS. Nevertheless, particles containing primary components are present within the ATAL (Figs. 8b and 9). However, these particles also contain secondary material similar to the purely secondary type 1 (Fig. 8b) and contribute only a minor fraction to the number of analyzed particles between 370 and $400\,\mathrm{K}$ (Fig. 9b). Together, the external mixing of the secondary particle type 1 and the mixed





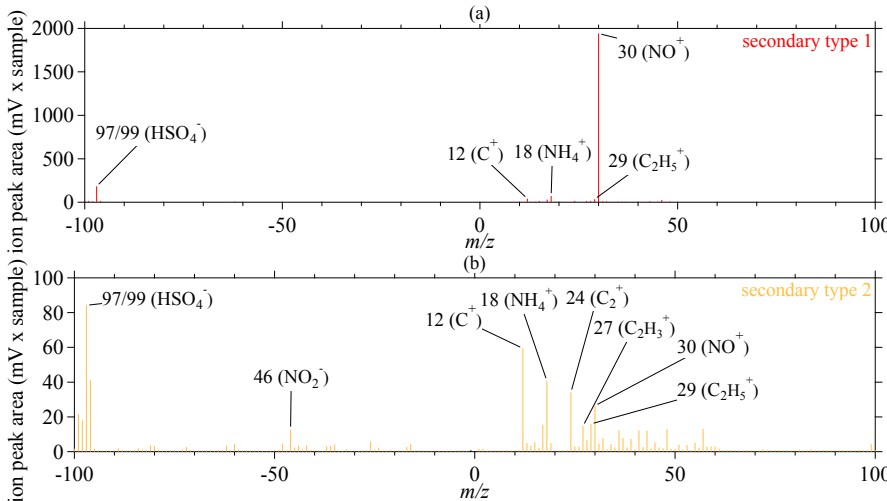

**Figure 7.** Bipolar mean spectra of the ERICA-LAMS particle types: (a) Secondary type 1 (averaged over 33420 particles), (b) Secondary type 2 (averaged over 6219 particles).

type suggests that the dominant fraction of particles within the ATAL is formed via secondary processes, hence, independent of pre-existing transported primary particles. Consistent with this result, the size distribution of the secondary type 1 exhibits smaller diameters compared to the distribution of primary particles in the ATAL (Fig. 10).

The single particle analysis of secondary particle type 1 shows the presence of particulate organics in the ATAL (Fig. 7a). By using the ERICA-AMS measurements, we further investigate whether these organics result from primary emissions or sec-

ondary formation. For each data point, the fraction of the organic signal at $m/z$ +44 to the total organic signal ($f_{44}$) is plotted against the fraction of organic $m/z$ +43 to total organic signal ($f_{43}$) in Fig. 11. The relationship between $f_{44}$ and $f_{43}$ is a proxy for the degree of oxidation of organic aerosol particles (see Supplement Sect. S1.6 and Ng et al. (2010)). We averaged the data over 90 s and only displayed values where the total organic signal as well as the organic signals at $m/z$ +43 and +44 are above the detection limit, since fractions of numbers below the limit of detection are very noise sensitive and contain no information.

Figure 11 additionally shows median and interquartile ranges of $f_{44}$ and $f_{43}$ for different altitude regimes (BL, lower ATAL, middle ATAL, upper ATAL). The organic aerosol within the ATAL shows a large variety in values of $f_{44}$ and $f_{43}$, which indicates the presence of aerosol particles with a wide range of ageing-states and/or from different sources (Ng et al., 2010, 2011). However, our observations of high $f_{44}$ (> 0.05) and low $f_{43}$ values are consistent with expectations about secondary organics that have experienced oxidative ageing, as demonstrated by Ng et al. (2011). Together with the single particle measurements,

we thus have several indications suggesting that the ATAL composition is to a large part controlled by secondary organic and inorganic aerosol formation. Our findings complement concurrent in situ observations of particle microphysical properties by Mahnke et al. (2021) and Weigel et al. (2021b), demonstrating that new particle formation and subsequent particle growth by condensation and coagulation takes place in the lower ATAL region below the tropopause (see Fig. 3).



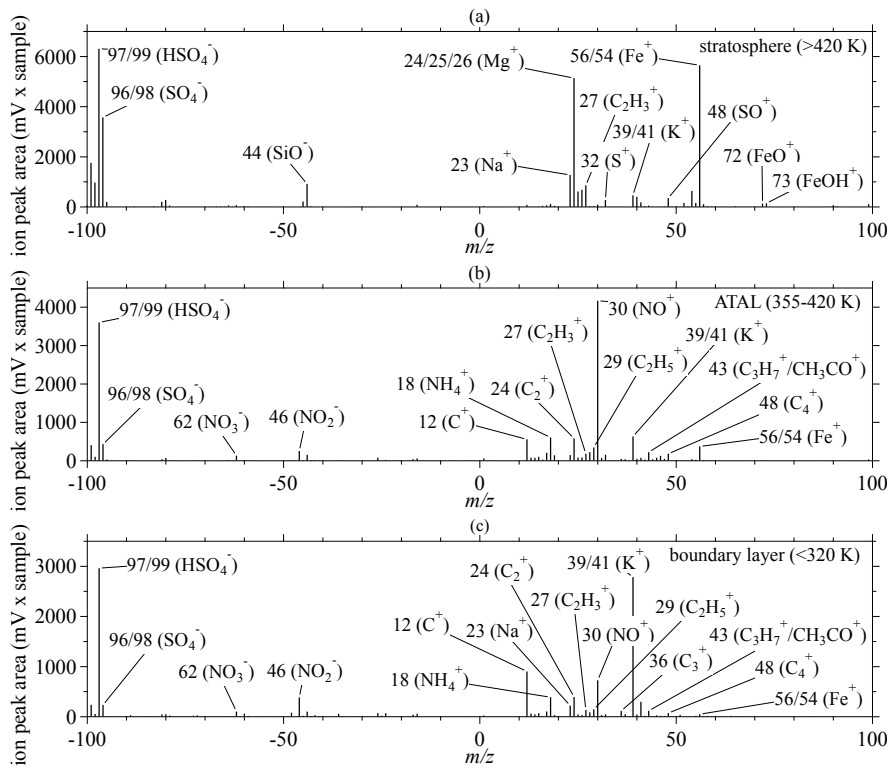

**Figure 8.** Bipolar mean spectra of the ERICA-LAMS particles containing primary components (mixed type): (a) in the stratosphere (averaged over 22493 particles), (b) in the ATAL (averaged over 16061 particles), and (c) in the boundary layer (averaged over 12044 particles).

The chemical nature of the ATAL has been subject of controversial discussions in the past decade. Modelling and observational
(including satellites) studies provided evidence that main constituents are sulfates, organics, nitrates, and ammonium (Fadnavis et al., 2013; Neely III et al., 2014; Yu et al., 2015; Gu et al., 2016; Lelieveld et al., 2018; Fairlie et al., 2020), while others discussed the importance of mineral dust and black carbon aerosol (Fadnavis et al., 2013; Lau et al., 2018; Brühl et al., 2018; Ma et al., 2019; Yuan et al., 2019; Bossolasco et al., 2021). Our in situ observations of particle composition, including primary and secondary components, show for the first time that the ATAL is largely controlled by secondary formation of nitrate, am-
monium, and organic aerosol, as proven by the predominance of purely secondary particles (by the ERICA-LAMS) as well as of ammonium nitrate and organics (by the ERICA-AMS). Nevertheless, the abundance of potassium, iron, and elemental carbon in the mean spectrum of the primary ATAL aerosol (Fig. 8b) indicates that primary emitted BL particles, likely from biomass burning (e.g. Silva et al., 1999; Pratt et al., 2011), anthropogenic emissions (e.g. Guazzotti et al., 2003), and mineral dust (e.g. Schmidt et al., 2017) can also be transported within the AMA into the tropopause region (e.g. Hudson et al., 2004;
Schill et al., 2020). In addition, the presence of iron in the primary ATAL aerosol particles (Fig. 8b) mainly originates from meteoric material that is transported from above into the upper ATAL region (Schneider et al., 2021). A detailed analysis of

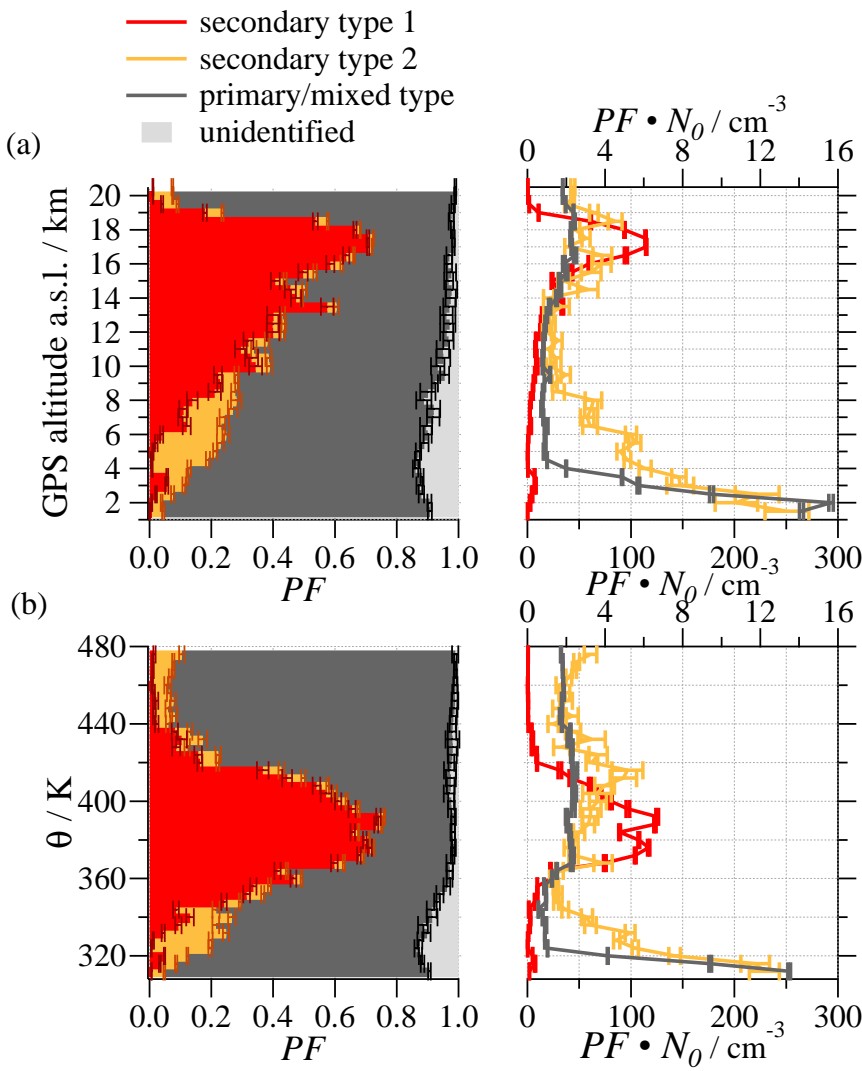

**Figure 9.** Vertically resolved fraction and number concentration of the identified particle types measured by the ERICA-LAMS as a function of (a) GPS altitude and (b) potential temperature $\theta$. The identified particle types are: secondary type 1 (red), secondary type 2 (orange), and primary or mixed type (dark grey). The unidentified particles are given in light grey in the left panels (see Supplement Sect. S2.1). The left panels represent the cumulative particle fractions ($PF$) of the particle types. The right panels represent the scaled ERICA-LAMS number concentrations ($PF \cdot N_0$) of the particle types. In the right panels, the top axis refers to the secondary particle type 2 and the bottom axis refers to the secondary particle type 1 and the primary or mixed type. Uncertainty analysis is given in the Supplement Sect. S2.2.





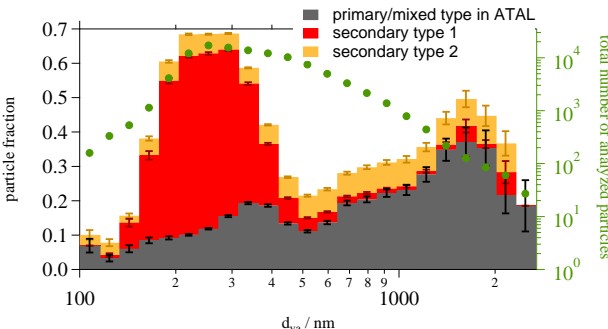

**Figure 10.** Size-resolved fraction of the identified particle types as measured by the ERICA-LAMS. The figure represents the cumulative particle fraction of the identified particle types: secondary type 1 (red) and secondary type 2 (orange), as well as the total number of analyzed particles per bin (dark green). The distribution of primary type particles (dark grey) is limited to particles in the ATAL. Uncertainty analysis is provided in the Sect. S2.2. This graph does not show the complete size distribution of the ambient aerosol, especially the lower cutoff is strongly influenced by the ability to detect small particles with ERICA.

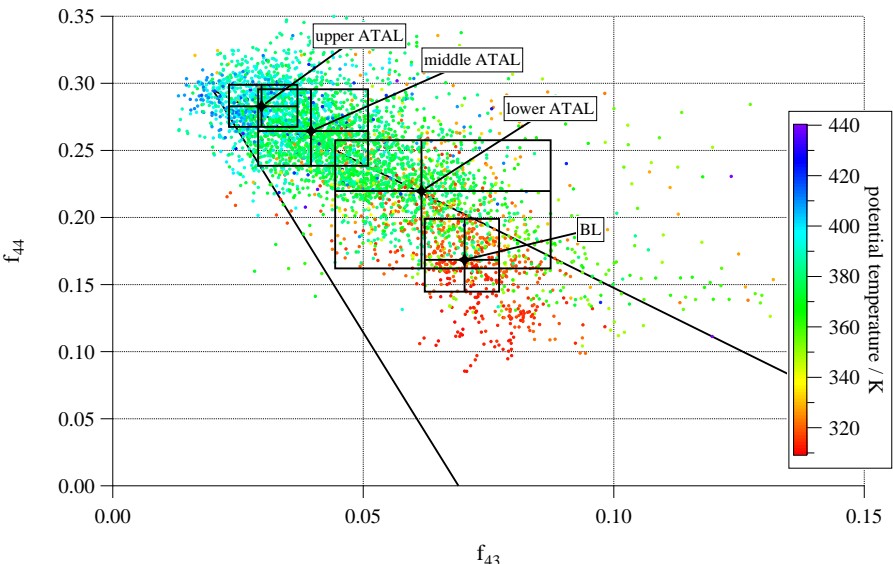

**Figure 11.** Graph showing the ratios $f_{44}$ vs. $f_{43}$ as indication for the degree of oxidation of the organic aerosol (colour coded with respect to potential temperature). Among all data points, where the total organic signal as well as the organic signals at $m/z$ 43 and 44 are above the detection limit, we present the median values for the boundary layer ($\theta < 320\,\mathrm{K}$), the lower ATAL (355 K-370 K), the middle ATAL (370 K-390 K), and the upper ATAL (390 K-420 K) in a box representing the corresponding $25th$ and $75th$ percentiles. For the median values all data points, where organics are above the detection limit are considered. Black lines show the triangular area according to the criteria introduced by Ng et al. (2010).





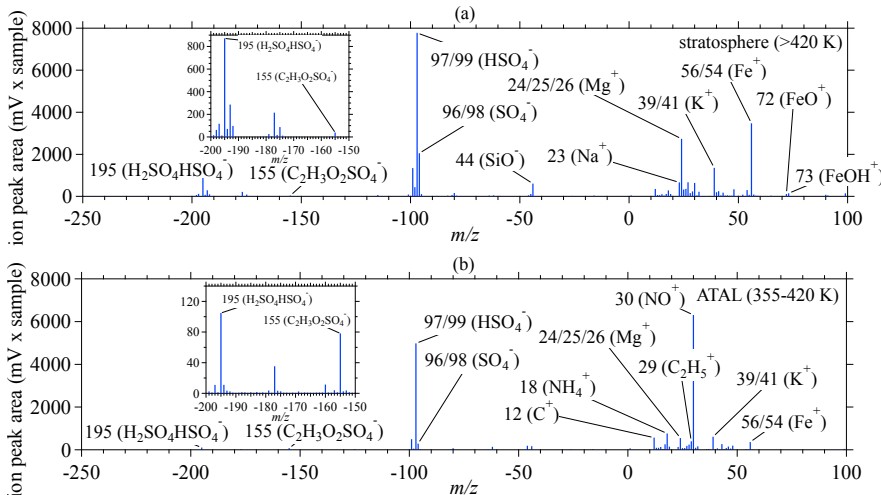

**Figure 12.** Bipolar mean spectra of the ERICA-LAMS particles containing organosulfates (here: glyoxal or glycolic acid sulfate evident by the abundance of $m/z$ -155 in single particle spectra) in (a) the stratosphere above 420 K potential temperature (averaged over 2354 particles) and (b) the ATAL between 355 and 420 K (averaged over 11176 particles).

the measured primary particles and its vertical uplift within the AMA is subject of a follow-up study.

### 3.4 Measurements of particulate organosulfates

To gain further insight into the composition of secondary organic aerosol (SOA) within the ATAL, our single particle mea-
surements by the ERICA-LAMS can be used as well. According to earlier studies, the LDI technique is able to detect the
molecular identity of some organic compounds, as for example of organosulfates (OS), which are known as tracers for SOA
(Froyd et al., 2010; Hatch et al., 2011a, b; Liao et al., 2015). During the StratoClim aircraft campaign 2017, we observed a
significant fraction of particles containing OS in the ATAL and even in the lower stratosphere. The most abundant marker ion
in the single particle spectra is given by $m/z$ -155 ($C_2H_3O_2SO_4^-$), suggesting the presence of glyoxal sulfate or glycolic acid
sulfate (GA sulfate; Froyd et al., 2010; Surratt et al., 2008; Galloway et al., 2009; Hatch et al., 2011a; Liao et al., 2015). The
isotopic ratio of the $m/z$ -155 and $m/z$ -157 signal of 0.06 is in line with the isotopic abundance of the sulfur (and oxygen)
isotopes. We found that 21 % of all analyzed particles by number contain GA sulfate. The mean bipolar spectra of this particle
type in the stratosphere and in the ATAL are given in Fig. 12. Within the ATAL, GA sulfate is largely internally mixed with
ammonium and nitrate (Fig. 12b). Whereas in the stratosphere, GA sulfate is found on particles containing meteoric material
and sulfuric acid (Fig. 12a; Schneider et al., 2021).

Figure 13 shows the vertical profile of number fraction and scaled number concentration $PF \cdot N_0$ of particles containing GA
sulfate identified by a peak at $m/z$ -155. These particles are largely present in the ATAL, particularly in the upper ATAL re-



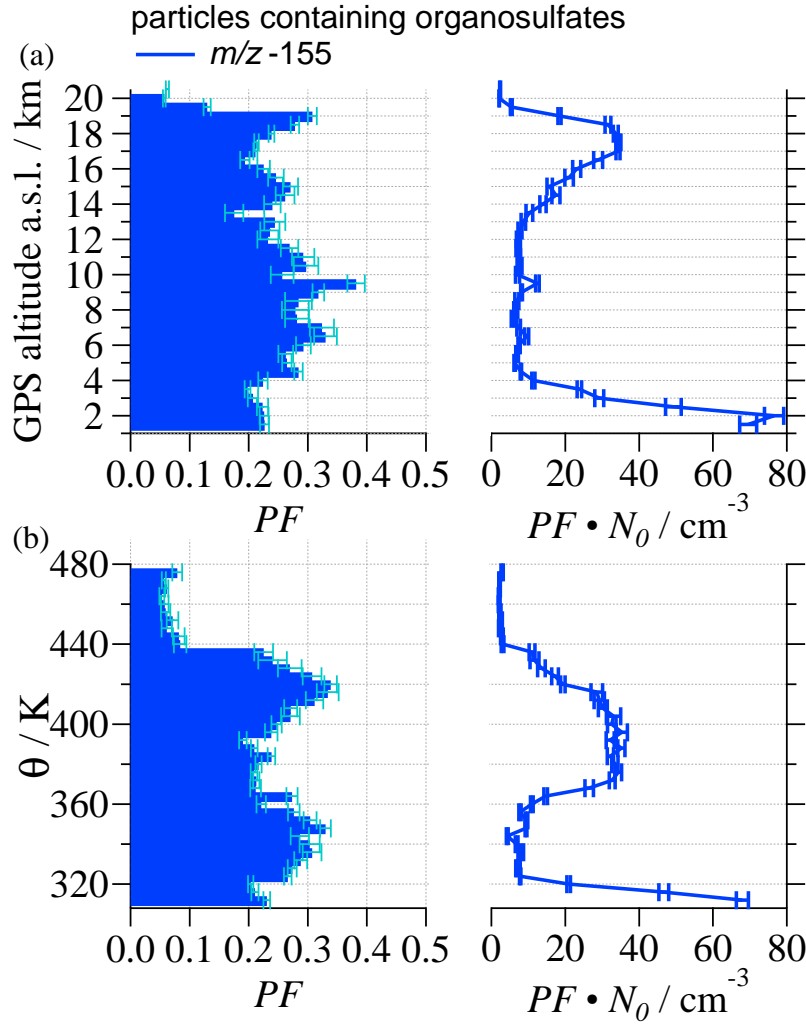

**Figure 13.** Vertically resolved fraction and number concentration of the particles containing organosulfates (here: glyoxal or glycolic acid sulfate evident by the abundance of $m/z$ -155 in single particle spectra) detected by the ERICA-LAMS as a function of (a) GPS altitude and (b) potential temperature $\theta$. The left panels represent the particle fraction ($PF$) of the particle type. The right panels represent the scaled ERICA-LAMS number concentrations ($PF \cdot N_0$). Uncertainty analysis is given in the Supplement Sect. S2.2.





gion. The formation of GA sulfate is dependent on the availability of precursor gases, $NO_x$ levels above 1 ppb (Surratt et al.,
2008; Froyd et al., 2010; Wennberg, 2013), and the formation of GA sulfate is enhanced at high aerosol acidity (Froyd et al.,
2010). Consistently, medium to high volume mixing ratios of NO (as proxy for $NO_x$) were observed within the ATAL, ranging
between 0.5 and 5 ppbv with maximum values of up to 20 ppbv in very fresh convective outflow (Stratmann et al., 2021).
Comparing Figs. 6 and 13, we can further conclude that the number fraction of particles containing GA sulfate increases with
increasing aerosol acidity above 400 K (above 18 km). In line with this is the observation that the ERICA-LAMS spectra of the
GA sulfate particles also concurrently exhibit ion peaks at $m/z$ -195 and -97 ($H(HSO_4)_2^-$ and $HSO_4^-$, respectively), which is
indicative for the presence of acidic sulfate in the single particles (Yao et al., 2011). Gas-phase organic precursors of GA sulfate
are diverse and as such subject of various laboratory and modelling studies (e.g. Surratt et al., 2008; Galloway et al., 2009;
Schindelka et al., 2013; Liao et al., 2015; Brüggemann et al., 2020). The authors discuss glyoxal, glycolic acid, and methyl
vinyl ketone together with their precursors (e.g. isoprene, ethene, acetic acid, acetylene, and aromatic compounds) from anthro-
pogenic and biogenic sources as potential candidates. However, such gas-phase organic precursors were not measured during
the StratoClim 2017 aircraft campaign, thus we could not provide further information on the availability and identification of
precursor gases to form GA sulfate. In summary, we have indications that the abundance of SOA within the ATAL can be partly
attributed to the formation of organosulfates in this region.

### 3.5 Photochemical processing of organic aerosol within the ATAL and the lower stratosphere

Particle composition in the ATAL is further influenced by photochemical processing of aerosol particles, along with the vertical
uplift of air masses even reaching across the tropopause. By using the ERICA-AMS measurements, we calculated the ratio
$R_{44/43}$ as the fraction of $f_{44}$ to $f_{43}$ to show the degree of oxidation of the organic aerosol as a function of altitude (Fig. 14; see
Supplement Sect. S1.6 for further details). In the BL and lower troposphere, the ratio is relatively constant. Within the ATAL
the ratio is significantly increasing with altitude and potential temperature, demonstrating the increasing degree of oxidative
ageing of organic species (Fig. 14). We thus conclude that particles formed in the lower ATAL region are further uplifted
and thereby subject to extensive oxidative ageing until a potential temperature of 440 K. The ensuing vertical transport of air
masses above the Asian monsoon had been earlier demonstrated by several modelling studies (e.g. Pan et al., 2016; Ploeger
et al., 2017; Vogel et al., 2019). Above the main convective outflow of the AMA ($\sim$ 360 K) radiative heating is the main process
driving vertical transport until $\sim$ 460 K with about 1 K per day - 1.5 K per day (Ploeger et al., 2017; Vogel et al., 2019). Along
with the vertical uplift air inside the AMA, we have indications that SOA formation is ongoing in the lower stratosphere.
At these altitudes, stratospheric background air gains in influence and mixes with the uplifted AMA air (Vogel et al., 2019).
This dilution process results in decreasing concentrations of ammonium, nitrate, and organic aerosol particles. However, we
observed that the mass fraction and mass concentration of organic aerosol shows a less steep decrease towards altitudes above
18 km or 420 K compared to nitrate (Fig. 15 and 2). This might be explained by ongoing SOA formation, competing with
the dilution process and in turn extending the vertical profile of organic matter towards higher altitudes compared to nitrate.
Apparently volatile organic precursors are still available, while the precursor for ammonium nitrate, i.e. ammonia from ground
emissions, is not available in relevant concentrations.



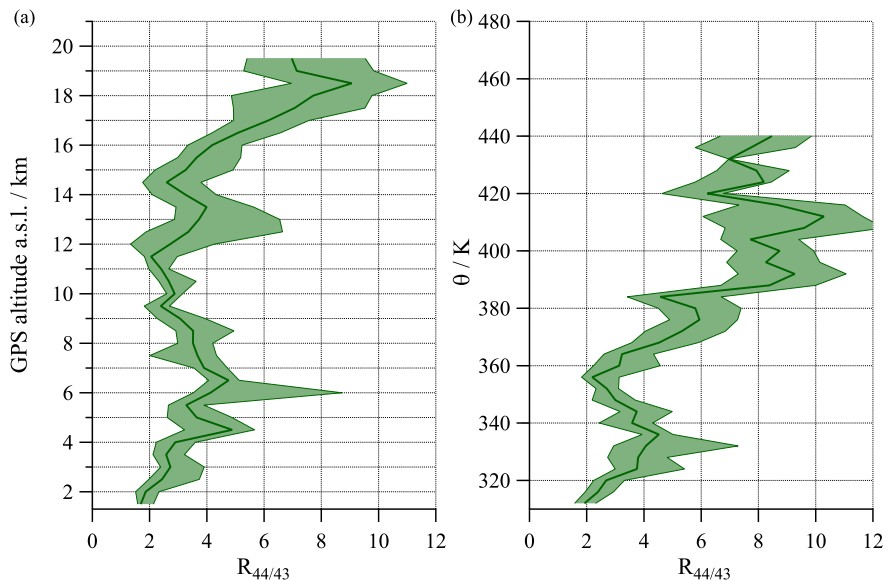

**Figure 14.** Vertical profiles with median and interquartile ranges of the fraction $R_{44/43}$ derived from the ERICA-AMS measurements as a function of (a) GPS altitude and (b) potential temperature $\theta$. Values above 18.5 km or 420 K are uncertain due to low organic signals, which are very sensitive to $CO_2$ corrections.

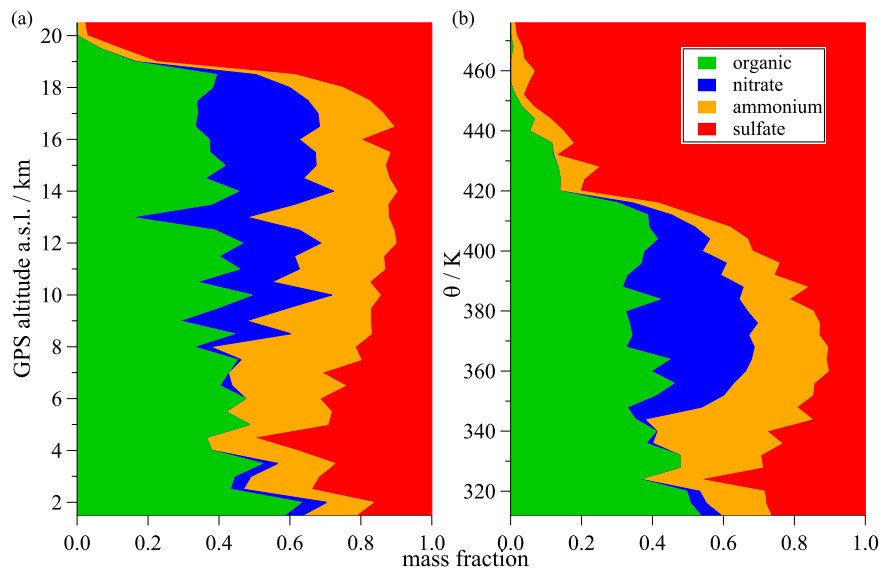

**Figure 15.** Vertical profiles of the cumulative mass fraction of particulate organics (green), nitrate (blue), sulfate (red), and ammonium (yellow) from the ERICA-AMS as a function of (a) GPS altitude and (b) potential temperature $\theta$.





Consistent with this, we observed the presence of GA sulfate in the lower stratosphere, resulting in a maximum fraction of more than 30 % of the particles to contain GA sulfate at 420 K (Fig. 13). This is in line with earlier studies demonstrating that

the formation of GA sulfate is supported by high aerosol acidity as found in high altitudes (Liao et al., 2015). The maximum fraction of GA sulfate containing particles at 420 K (Fig. 13) is likely formed when precursor gases from below encounter highly acidic sulfate particles from the stratosphere (Liao et al., 2015). However, the abundance of GA sulfate particles is decreasing in the lower stratosphere, which implies that the formation of GA sulfate is at a certain point also limited by the amount of gas phase precursors from below, rather than aerosol acidity. Together, we conclude that organic aerosol in the lower

stratosphere above the AMA is partly controlled by secondary formation of organosulfates that in turn is driven by the presence of stratospheric acidic aerosol and the availability of precursor gases from the AMA uplift.

## 4    Conclusions

Motivated by the limited knowledge of chemical particle composition within the ATAL, we conducted in situ aerosol mass spectrometric measurements at high altitudes (up to 20 km or 480 K in potential temperature) in the summertime AMA. The

novel, hybrid aerosol mass spectrometer ERICA was deployed on the Russian high altitude research aircraft M-55 *Geophysica* for eight research flights during the StratoClim field campaign based in Kathmandu, Nepal, from July 27 until August 10, 2017. The presence of the aerosol layer with increased concentrations of particulate nitrate, ammonium and organics in the altitude range between 11 km and 19 km (corresponding to ~355 K and 420 K) could be demonstrated by means of the mass spectrometric in situ measurements. The layer thus ranges from below to well above the thermal tropopause, which can be

found at about 380 K. For organics, concentrations above the detection limit are found even at potential temperatures slightly higher than 420 K. The vertical profile of particulate sulfate did not exhibit a layered structure as the sulfate mass concentration increased steadily with altitude up to 20 km. The existence of the ATAL also was evident in the same altitude range from the concurrent measurements of the condensation particle counter COPAS (detecting particles larger than ~10 nm) as well as the optical particle counter UHSAS-A (for particles larger than ~65 nm). Apparently, nitrate and organics are the dominant non-

refractory components of the ATAL. For both components a concentration of around $0.8\,\mu g\,m^{-3}$(NTP) were found as median values over the whole campaign. The organics play a comparable role to nitrate for the ATAL formation and persistence. Analyses of the concurrent availability of ammonium, nitrate, and sulfate showed that particles found up to an altitude of 18 km (~400 K) contain sufficient ammonium for completely neutralizing sulfate and nitrate. Up to 19 km (~420 K) ammonium was still detected, but not in sufficient amounts for full neutralization. However, in these analyses the possible existence of organic

acids could not be considered due to the lack of suitable measurements.

Regarding secondary aerosol formation, three main pieces of evidence prove an important role of organic and inorganic secondary aerosol formation for the particle composition in the ATAL. First, by means of the LDI technique (the ERICA-LAMS), we found a dominant fraction (up to 70 % by number) of purely secondary particles particularly in the ATAL. Those particles were characterized by an internal mixture of nitrate, ammonium, sulfate, and organic matter; and as such are externally

mixed with primary components (e.g. potassium and elemental carbon). Second, the size distribution of these secondary type




particles shows lower values compared to particles containing primary components in the ATAL. Third, our observations by the ERICA-AMS show medium to high values of $f_{44}$ ($> 0.05$) within the ATAL, indicating the predominance of secondary organics that have experienced oxidative ageing, rather than of fresh primary organics. Moreover, we found that organic matter in the ATAL can partly be identified by the ERICA-LAMS as organosulfates, known tracers for secondary organic aerosol. Par-

ticles containing glyoxal or glycolic acid sulfate (GA sulfate) show a maximum fraction in the upper ATAL region and lower stratosphere (380 - 420 K). This result might be explained by the uplift of organic precursor gases from below that encounter stratospheric acidic sulfate aerosol. The abundance of particles containing GA sulfate in the upper ATAL coincides with an increase in aerosol acidity (obtained from ERICA-AMS measurements). Consistent with this, the single particle mass spectra show an internal mixture of GA sulfate with acidic sulfate partly associated with material that can be identified as meteoric

dust. Furthermore, we occasionally observed medium to high NO-regimes ($> 1$ ppb with maximum values of 20 ppb) in the ATAL, which is in line with results from earlier studies on the formation of GA sulfate. We showed that the degree of oxidation of organic aerosol measured by the ERICA-AMS increases with altitude and potential temperature above 12 km. In particular the measurements show that the particles in the upper ATAL and the lower stratosphere reached higher degrees of oxidation than in the lower ATAL. This indicates that particles formed in the lower ATAL are uplifted and thereby exposed to extensive

oxidative ageing, consistent with earlier studies demonstrating the slow vertical uplift of AMA air in an anticyclonic spiralling range from $\sim 360$ K up to $\sim 460$ K (Ploeger et al., 2017; Vogel et al., 2019).

Inherent in the adopted experimental techniques, which rely on the detection of relatively small molecule fragments, is the well known disadvantage of not being able to unambiguously identify individual particle components (especially organic species). This is an important task for future instrumental development. Also, the data set from these eight research flights represents

merely a fairly short period within only one AMA season. Further detailed in situ measurements inside the AMA are needed, especially also from the side of the Tibetan plateau, which appears as exceedingly difficult considering the current geopolitical situation. Also dedicated research on the processing of the aerosol and trace gases in the outflow regions of the AMA is necessary to assess the effects on radiation and cloud formation and the longer term implications of the aerosol generated within the AMA. Another open question is, how much of the aerosol actually reaches the tropical pipe entering further vertical uplift

or even the stratospheric Junge layer. Taken together, our experiments provided first details on the chemical composition of the ATAL especially with respect to secondary aerosol, and the results indicate the important role of secondary organic and inorganic particle formation for the presence of the ATAL.

*Data availability.* The ERICA mass spectrometry data will be available in the Edmond database (Edm, 2017) and the Halo database (Hal, 2017).

*Author contributions.* OA performed the ERICA-AMS analysis and generated the corresponding graphs, FK performed the ERICA-LAMS analysis and generated the corresponding graphs. Instrument development and operation during the campaign was done by AD, SM, OA, AH,



and SB. RW and CM operated and evaluated the data of the COPAS and UHSAS-A, respectively. HS provided the NO data from SIOUX. FD and CS gave critical input for understanding and interpreting the ERICA-AMS data of the new ERICA instrument and provided help for solving instrumental issues. MP analyzed the cloud data and generated the cloud mask for data selection. FS was managing campaign
operation, especially designing the flight paths to achieve the scientific goals. BV provided expertise on the Asian monsoon dynamics. The manuscript was written by SB, FK, and OA. All authors commented on the manuscript.

*Competing interests.*  The authors declare no competing interests

*Acknowledgements.*  The StratoClim project was funded by the EU (FP7/2007–2018 Grant No. 603557) and supported by the German Federal Ministry of Education and Research (BMBF) under the ROMIC-project SPITFIRE (01LG1205A). The work on ERICA was financed by
the European Research Council under the EU's Seventh Framework Program (FP/2007-2013)/ERC Grant Agreement No.321040 ("EXCA-TRO") and supported by the Max Planck Society. The presented work includes contributions of the NSFC–DFG 2020 project ATAL-track (BO 1829/12-1 and VO 1276/6-1). We thank the workshops of the Max Planck Institute for Chemistry and of the Institute for Physics of the Atmosphere (Mainz University) as well as T. Böttger for building and implementing our instruments on the M-55 *Geophysica*, and Anneke M. Batenburg (now at Platform Talent voor Technologie, the Hague, Netherlands) for her valuable help during the field campaign and data
collection. Also, we gratefully acknowledge Tofwerk AG (Switzerland) for their essential support, in particular M. Cubison for customizing "Tofware". Felix Plöger (IEK-7, Forschungszentrum Jülich GmbH, Jülich Germany) provided the equivalent latitude data which we gratefully acknowledge. We would like to express our gratitude to M. Rex (Alfred Wegener Institut – Helmholtz Zentrum für Polar und Meeresforschung) for managing the entire StratoClim project. We gratefully acknowledge the crew of MDB (Myasishchev Design Bureau) and the M-55 *Geophysica* pilots for their engagement and dedication. Finally, we extend our most sincere thanks to Nepalese government au-
thorities, involved research institutions, and Tribhuvan Airport, as well as the German Embassy, for their outstanding, extraordinary support, and their fabulous hospitality that enabled our field campaign and research.





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
