# Peer review of "S1 Signal processing and data analysis for the ERICA-AMS"

_Atmospheric Chemistry and Physics, 2022_

## Author Comment (AC1)

**Chemical analysis of the Asian Tropopause Aerosol Layer (ATAL) with emphasis on secondary aerosol particles using aircraft based in situ aerosol mass spectrometry**

Appel et al.
**Replies to the comments by Anonymous Referee #1**

General Reply:
We would like to thank Referee #1 for her/his encouraging, constructive comments which helped us to improve the manuscript. In the following, we comment on the individual points.

The reviewer comments are written in black.

Our answers are written in blue.

Changes to the revised version of the manuscript are printed in red.

**Review of "Chemical analysis of the Asian Tropopause Aerosol Layer (ATAL) with emphasis on secondary aerosol particles using aircraft based in situ aerosol mass spectrometry"**

Appel et al. discuss the results of aerosol in-situ observations taken on board a high-altitude aircraft within the European Union's project StratoClim from Kathmandu/Nepal in July and August 2017. The paper concentrates mainly on particle composition within the Asian Tropopause Aerosol Layer (ATAL) probed with an aerosol mass spectrometer. It is shown that the aerosol particles of the ATAL as observed during the flights were mainly composed of ammonium nitrate and organics. While ammonium nitrate as a major component of the ATAL has already been reported in a previous publication, the role of organics is an important new finding of this work. Moreover, the authors present convincing arguments for the importance of secondary particle formation for the majority of the aerosols observed within the ATAL.

The manuscript provides an extremely important contribution to our understanding of an (concerning in-situ measurements) under-sampled region of the Earth's atmosphere – also given the uncertainties connected to aerosols and their radiative direct and indirect impacts within the climate system. The analysis of the measurements and their interpretation is convincing. The paper is well written containing clear graphical illustrations. Thus, I strongly support its publication after taking into consideration some minor comments below and after making available the datasets ('Data availability').

Specific comments:

L69: 'from the CRISTA instrument'

Many of the data in the quoted publication stem also from the MIPAS/Envisat instrument.

Has been changed to:

from the instruments CRISTA and MIPAS/Envisat

L176: 'During the StratoClim aircraft campaign 2017 we found an average detection limit of 0.12 µgm−3'

Please add '(NTP)' where applicable.

Done.

Figure 2:

It would be good to show (perhaps in the supplement) the number of data points per altitude bin, since lower altitudes are only sampled during take-off and landing while much more measurements exist at flight level.

The amount of data points influences the statistical significance of the measurements in each bin. This significance is reflected by the corresponding detection limit (dotted line in Fig. 2). The number of points per altitude/Theta bin is given the supplementary data files. We included a corresponding graph in the supplement following this reviewer's suggestion.

L298: 'in the ATAL is predominantly existent in the form of ammonium nitrate (AN)'

Shouldn't ammonium sulfate also be present and (at the top of the ATAL) be of similar importance?

Yes, that is correct regarding the quantitative abundance. As written in lines 274ff sulfate becomes more dominant with higher altitude, but does not show a layered feature as required to explain the layered structure of the ATAL. The neutralization plot suggests that most of the sulfate is ammonium sulfate, especially up to 390 K. We implemented the following text in line 275:

(mostly as ammonium sulfate below 400 K potential temperature, see Fig. 6)

L317 and Fig. 4: 'Nevertheless, the data show that the ATAL for the time of StratoClim was not only confined in the vertical direction but also indicates a decrease towards the edge of the AMA in the horizontal distribution.'

It is not very clear to me where to see in Fig. 4 the horizontal confinement in the altitude region of the ATAL. There might be a slight decrease in organics, however in nitrate, the data of which reaches to much lower equivalent latitudes, I cannot see a clear decrease.

The decrease might be to faint to conclude a horizontal confinement. We reconsidered the statement and changed the text to:

A clear horizontal confinement of the ATAL can not be discerned from the graph. This might have been different, if the data base would be larger.

L321: 'Stratmann et al., 2021'

This paper has not been published. Is it available elsewhere?

As the paper has not been published in the meantime, we deleted this reference. We added instead Lelieveld et al. (2018). In other cases, we added "as measured by SIOUX" in agreement with Hans Schlager, who is a co-author of the manuscript.

L324, Figure 5:

It is not clear what this Figure should tell: NO is steadily increasing with altitude and nitrate has a maximum within the ATAL. Can any more information be derived from the regression lines?

The referee describes well the behaviour of the displayed quantities. The common increase of NO and nitrate mass concentration at altitudes between 360 and 380 K (i.e. at altitudes where cloud outflow still exists) indicates that lightning NOx is available for ammonium nitrate formation. At higher altitudes the nitrate aerosol decreases despite of the abundance of the precursor NOx. This indicates a lack of the other necessary precursor for particulate nitrate formation, NH3, which also is documented by the GLORIA/MIPAS measurements shown in Höpfner et al, 2019. The following text was added in lines 459-466:

This hypothesis is supported by GLORIA measurements of the precursor gases acetylene and ammonia (Johansson et al., 2020; Höpfner et al., 2019, respectively) and measurements of the precursor gas NO (Fig. 5). Observations show that acetylene and NO can reach the altitude of the tropopause and above (Fig. 2g in Johansson et al., 2020), while ammonia as precursor gas of ammonium nitrate shows enhanced concentrations at a maximum altitude of 15 km (Fig. 3d in Höpfner et al., 2019). Thus, organic precursor gases are still available even above the tropopause, while ammonia is not present in relevant concentrations. A possible explanation for the difference in the vertical distribution of acetylene and ammonia can be found in their lifetime. Ammonia reacts fast with sulfuric acid or nitric acid, whereby acetylene has a lifetime of about 2 weeks (Johansson et al., 2020).

Johansson, S., et al.: Pollution trace gas distributions and their transport in the Asian monsoon upper troposphere and lowermost stratosphere during the StratoClim campaign 2017, Atmos. Chem. Phys., 20, 14695–14715, https://doi.org/10.5194/acp-20-14695-2020, 2020.

Höpfner, M., et al.: Ammonium nitrate particles formed in upper troposphere from ground ammonia sources during Asian monsoons, Nature Geoscience, 12, 608–612, https://doi.org/10.1038/s41561-019-0385-8, 2019.

The regression lines do not provide further information. We intended to highlight the trend in the combined data sets.

L404-416:

This paragraph seem a bit detached at its current position. May to be possible to transfer it into the 'Conclusions'?

We agree. We transferred a part of this text into the conclusions.

L465: 'Apparently volatile organic precursors are still available, while the precursor for ammonium nitrate, i.e. ammonia from ground emissions, is not available in relevant concentrations.'

This conclusion seems rather indirect. Are there any independent measurements of the precursor gases to support it? Furthermore, what might be the reason for it?

We agree. This statement is rather indirect and speculative, so far. According to your suggestion, we looked into the GLORIA measurements of acetylene (organic precursor gas) and ammonia (precursor gas of ammonium nitrate and ammonium sulfate) on July 31 (Höpfner et al., 2019 and Johansson et al., 2020, respectively). Comparing the respective vertical profiles (Fig. 3d in Höpfner et al., 2019 and Fig. 2g in Johansson et al., 2020), it is obvious that acetylene can reach altitudes of the tropopause and above, while ammonia shows enhanced values at a maximum altitude of 15 km.

For this reason, we re-worded this part as follows:

This hypothesis is supported by GLORIA measurements of the precursor gases acetylene and ammonia (Johansson et al., 2020; Höpfner et al., 2019, respectively) and measurements of the precursor gas NO (Fig. 5). **Observations show that acetylene and NO** can reach the altitude of the tropopause and above (Fig. 2g in Johansson et al., 2020), **while ammonia** as precursor gas of ammonium nitrate shows enhanced concentrations at a maximum altitude of 15 km (Fig. 3d in Höpfner et al., 2019). Thus, organic precursor gases are still available even above the tropopause, while ammonia is not present in relevant concentrations. A possible explanation for the difference in the vertical distribution of acetylene and ammonia can be found in their lifetime. Ammonia reacts fast with sulfuric acid or nitric acid, whereby acetylene has a lifetime of about 2 weeks (Johansson et al., 2020).

Johansson, S., et al.: Pollution trace gas distributions and their transport in the Asian monsoon upper troposphere and lowermost stratosphere during the StratoClim campaign 2017, Atmos. Chem. Phys., 20, 14695–14715, https://doi.org/10.5194/acp-20-14695-2020, 2020.

Höpfner, M., et al.: Ammonium nitrate particles formed in upper troposphere from ground ammonia sources during Asian monsoons, Nature Geoscience, 12, 608–612, https://doi.org/10.1038/s41561-019-0385-8, 2019.

L521: 'which appears as exceedingly difficult considering the current geopolitical situation'

I don't think that this statement provides any information within the scope of the manuscript.

The stated unfortunate fact underlines the importance of further measurements, which reach beyond our data. It is also a reason for why aerosol data in this region are sparse. Such a statement helps scientists from the region of the Asian monsoon to write their own corresponding proposals and Coauthor SB was already contacted along these lines regarding the importance of desert dust. However, we changed the statement to read like this:

Further detailed in situ measurements inside the AMA are needed, especially also from the side of the Tibetan plateau.

L528: 'The ERICA mass spectrometry data will be available in the Edmond database (Edm, 2017) and the Halo database (Hal,2017).'

Is the dataset available now?

After addressing technical issues, the data are available at the EDMOND database since 10.03.2022. The data are also uploaded to the HALO database. The mission PI will make the data on the HALO database publicly accessible at some point in the future. The citations of the databases have been re-worded.

The ERICA mass spectrometry data are available from the Edmond database (Edmond, 2022) and will be additionally available from the HALO database (HALO-DB) (HALO-DB, 2017) in the near future.

Technical comments:

L4 (and elsewhere): 'high altitude' -> 'high-altitude'

Has been changed throughout the text.

L13 (and elsewhere): 'analyzed' vs. 'vapour' (L163) : AE and BE seem to be mixed in the manuscript

We all are non-native English speakers.To our understanding 'analyzed' can be written with 'z' in BE as well, e.g. in Oxford English. The ACP editorial services will help us with providing a consistent spelling throughout the manuscript during the step of proofreading and typesetting.

L27: Comma after 'As a consequence' missing; (also elsewhere: commas seem to be missing on similar expression at the beginning of a sentence)
Commas have been added to such phrases.

L206: 'sonde' -> 'probe'

Has been changed.

L219: '2021a, b, ,' -> '2021a, b, '

Has been changed.

L316: '(Fig. 4a)).' -> '(Fig. 4a).'

Has been changed.

**In addition to the changes in reply to the referees comments:**

Line 136: 20.5 km instead of 20.0 km. 20.0 km was the maximum pressure altitude reached, 20.5 km for GPS altitude.

Line 541: FP7/ instead of FP/

Fig.2: Caption has been changed from organic to organics and scale reduced to 2 µg/m^3 to improve the visibility of the enhancement in the ATAL.

We added to the caption of Fig. 2: Concentrations in the boundary layer can exceed the displayed range

Supplementary figures S2-S9: Caption has been changed from organic to organics.

In the introduction, one sentence is added introducing the new paper by Wang et al., 2022, which appeared on May 18, 2022. (Wang et al., Synergistic HNO3–H2SO4–NH3 upper tropospheric particle formation, https://doi.org/10.1038/s41586-022-04605-4, Nature, 2022)

According to the reviewers suggestions, we looked into the GLORIA measurements of acetylene on July 31 (Johansson et al., 2020). For this reason, we re-worded the sentence in lines 438-439 as follows:

The mixing ratios of acetylene measured with the GLORIA instrument feature increased values in the upper troposphere and even in the lower stratosphere (Johansson et al., 2020).

---

## Author Comment (AC2)

ACP-2022-92

**Chemical analysis of the Asian Tropopause Aerosol Layer (ATAL) with emphasis on secondary aerosol particles using aircraft based in situ aerosol mass spectrometry**

Appel et al.
**Replies to the comments by Anonymous Referee #2**

General Reply:
First of all, we would like to thank Referee #2 for the positive and constructive feedback. In the following, we will comment on the individual points.

The reviewer comments are written in black.

Our answers are written in blue.

Changes to the revised version of the manuscript are printed in red.

**Review of "Chemical analysis of the Asian Tropopause Aerosol Layer (ATAL) with emphasis on secondary aerosol particles using aircraft based in situ aerosol mass spectrometry"**

This paper discussed the unique aerosol and gas measurements aboard a high-altitude aircraft. It provided a promising dataset for atmospheric research, especially for understanding the aerosol particle composition within the Asian Tropopause Aerosol Layer (ATAL). The authors presented a very useful tool – ERICA-AMS for the atmospheric study and shared the exciting results from July and August 2017. The paper is well written. The topic is well aligned with the journal scope and should be considered for publication after minor revision.

Specific comments:

Page 9, lines 256-262. The discussion about Fig 2 in this section can not support this statement. "Consequently, the ATAL chemical composition is largely determined by the relative contributions of new particle formation and secondary particle growth at altitude compared to the upward transport of already nucleated secondary or of primary particles from below." Maybe include the gas phase measurements to indicate the new particle formation trend?

We agree. We removed this sentence.

Figure 2 showed the sulfate concentration increased from $0.5 – 1.5$ ug/m^3 above 19 km. However, Figure 3 showed that the particle number concentrations from COPAS or UHSAS were less than 90 #/cc. What does the size distribution above 19 km? Are those particles all

sulfuric acid? Even if we assume they were ammonium sulfate and larger than 110 nm, the integrated mass seemed still lower than the AMS data.

Yes, all these particles are sulfuric acid, or at least contain a coating of sulfuric acid. Our first analyses of the single particle data show that practically all negative ion spectra contain lines with sulfur fragments. (Pure sulfuric acid can not be detected by our 266 nm laser ablation system.) Thus, the single particle analysis and the acidity analysis show, that the particles above 19 km are mainly sulfuric acid containing particles. A juxtaposition of the AMS with the UHSAS data here is quite difficult, because: (1.) The detected size ranges of both instruments are not the same, especially for the large sizes. Also, the sampling inlets are quite different. (2.) The particle diameter definition of the UHSAS is optical diameter, calibrated with the refractive index of PSL. Thus the sizing of the UHSAS can deviate from particles with a different index, which is difficult to estimate because of the inclusions of primary material components in the particles. (3.) As mentioned by Mahnke et al. (2022), a custom-made pump system had to be integrated in order to operate at low pressures. There might be yet unquantified losses inherent in the inlet system. In order to resolve these issues dedicated laboratory experiments are planned for the future application of the UHSAS and ERICA during the ACCLIP campaign with HIAPER (2022) and the PHILEAS campaign (2023) with HALO.

Figure 5: are those data points are from the averaged data? If so, what is the uncertainty?

The data are the median values in individual altitude bins. We have included percentiles for nitrate and uncertainties for NO in the graph and added the following text to the caption:

The horizontal bars reflect the 25th to 75th percentile range for nitrate and the vertical bars present an uncertainty of 30 % for NO.

Section 3.3 provided essential information about the mixing state of the aerosol particles. The abstract also mentioned, "…the majority of the particles encountered in the ATAL consisted solely of secondary substances, namely an internal mixture of nitrate, ammonium, sulfate, and organic matter. These particles are externally mixed with particles containing primary components as well." Does the mixing state remain the same at different aerosol particle sizes? Do you see a spatial variance in the mixing state?

Figure 10 shows the size distributions of purely secondary particles (type 1 represents mainly ATAL particles) as well as the primary/mixed type, restricted to particles in the ATAL. While the secondary particles show the highest fraction at a vacuum aerodynamic diameter between 200 and 300 nm, particles containing primary material show a much wider distribution. Thus, the mixing state significantly changes with size, smaller particles being mainly secondary type, whereas larger particles often contain primary material.
A spatial variance was not observed in our analysis up to now. The results are being prepared for a forthcoming new manuscript. There we will pick up and address the reviewer's question/suggestion regarding a potential spatial inhomogeneity.

Figure 10, It is not clear to me what the particle fractions for each particle type are. For example, at 300 nm, primary was 15%, type 1 – 65% or 40%, type2 68% or 5%? Should they add up to 1?

The particle fractions are cumulative, thus in the example of 300 nm: ~15% primary inside the ATAL, ~45% type 1 and ~5% type 2. The fractions do not add up to one, because primary particles found outside the ATAL, as well as unidentified particles are not included. To clarify this we have included the following into the caption:

The distribution of primary particles is limited to particles in the ATAL, thus the fractions do not add up to one.

Figure 14, why is there no data above 19 km?

R44_43 includes a fraction of data for organics, org44 and org43. We only show data points, where all three values are above the calculated detection limit for the corresponding vertical bin. Due to the low values for organics concentration and thus org44 and org43, this is not the case for data above 19 km. In response the reviewer's question, we changed the caption to:

Only data points are shown where values of organic mass concentration as well as org44 and org43 signals are above the detection limit of the corresponding vertical bins. This condition is not fulfilled above 19 km and 440 K.

Line 528: 'The ERICA mass spectrometry data will be available in the Edmond database (Edm, 2017) and the Halo database (Hal,2017).' Should data be available by now?

After addressing technical issues, the data are available at the EDMOND database since 10.03.2022. The data are also uploaded to the HALO database. The mission PI will make the data on the HALO database publicly accessible at some point in the future. The citations of the databases have been re-worded.

The ERICA mass spectrometry data are available from the Edmond database (Edmond, 2022) and will be additionally available from the HALO database (HALO-DB) (HALO-DB, 2017) in the near future.

**In addition to the changes in reply to the referees comments:**

Line 136: 20.5 km instead of 20.0 km. 20.0 km was the maximum pressure altitude reached, 20.5 km for GPS altitude.

Line 541: FP7/ instead of FP/

Fig.2: Caption has been changed from organic to organics and scale reduced to 2 µg/m^3 to improve the visibility of the enhancement in the ATAL.

We added to the caption of Fig. 2: Concentrations in the boundary layer can exceed the displayed range.

Supplementary figures S2-S9: Caption has been changed from organic to organics.

In the introduction, one sentence is added introducing the new paper by Wang et al., 2022, which appeared on May 18, 2022. (Wang et al., Synergistic HNO3–H2SO4–NH3 upper tropospheric particle formation, https://doi.org/10.1038/s41586-022-04605-4, Nature, 2022)

According to the reviewers suggestions, we looked into the GLORIA measurements of acetylene on July 31 (Johansson et al., 2020). For this reason, we re-worded the sentence in lines 438-439 as follows:

The mixing ratios of acetylene measured with the GLORIA instrument feature increased values in the upper troposphere and even in the lower stratosphere (Johansson et al., 2020).